# Breaking through water-splitting bottlenecks over carbon nitride with fluorination

Ji Wu[1], Zhonghuan Liu[1], Xinyu Lin[1], Enhui Jiang[1], Shuai Zhang[1], Pengwei Huo[1], Yan Yan [1]✉, Peng Zhou [2]✉ & Yongsheng Yan[1]

Graphitic carbon nitride has long been considered incapable of splitting water molecules into hydrogen and oxygen without adding small molecule organics despite the fact that the visible-light response and proper band structure fulfills the proper energy requirements to evolve oxygen. Herein, through in-situ observations of a collective C = O bonding, we identify the long-hidden bottleneck of photocatalytic overall water splitting on a single-phased g-C$_3$N$_4$ catalyst via fluorination. As carbon sites are occupied with surface fluorine atoms, intermediate C=O bonding is vastly minimized on the surface and an order-of-magnitude improved H$_2$ evolution rate compared to the pristine g-C$_3$N$_4$ catalyst and continuous O$_2$ evolution is achieved. Density functional theory calculations suggest an optimized oxygen evolution reaction pathway on neighboring N atoms by C–F interaction, which effectively avoids the excessively strong C-O interaction or weak N-O interaction on the pristine g-C$_3$N$_4$.

Producing hydrogen from water splitting using particulate photocatalysts is a low-cost green technology for large-scale solar energy conversion[1–3]. As a two-dimensional metal-free inorganic nanomaterial, graphitic carbon nitride (g-C$_3$N$_4$) exhibits superior hydrogen generation ability from water splitting by adding small molecule organics as hole scavengers, with the hydrogen evolution rate under visible-light (≥420 nm) even higher than that of the commonly used titanium dioxide catalyst under ultraviolet illumination[4–6]. The overall water splitting has also been achieved on some g-C$_3$N$_4$-based composite photocatalysts by constructing in-plane or Z-scheme heterojunctions with oxygen evolution reaction (OER) active co-catalysts[7–9]. However, single-phased g-C$_3$N$_4$ was long considered incapable of photocatalytic overall water splitting due to the insufficient OER ability of g-C$_3$N$_4$ to directly produce and release oxygen from pure water[10,11]. Researchers have generally attributed this inadequate OER ability to the weak oxidation capacity of photo-induced valence band holes on g-C$_3$N$_4$, which led to the assertion that g-C$_3$N$_4$ as a photocatalyst was only active for hydrogen evolution[2,12].

Typically, the one-step excitation overall water splitting requires a semiconductor having a band gap larger than the thermodynamic requirement of 1.23 eV and spanning the redox potential of both HER

and OER[13,14]. However, with an energy band gap greater than 2.0 eV and the positions of both valence and conduction bands that fully meet the thermodynamic demands of water[15,16], single-phased pristine g-C$_3$N$_4$ catalysts still failed to directly extract O$_2$ from pure water, indicating that unknown factors rather than the high valence band position hinder the OER on g-C$_3$N$_4$. For single-phased g-C$_3$N$_4$, figuring out the bottleneck that hinders the OER and how to bypass such a bottleneck to achieve efficient overall water splitting under visible light is crucial.

Herein, by in situ observations using isotopic-labeled ($^{16}$O/$^{18}$O) diffuse reflection infrared Fourier transform spectroscopy (DRIFTS) and near-ambient pressure X-ray photoelectron spectroscopy (NAP-XPS), we confirmed that the accumulated C=O bonding at the H$_2$O/g-C$_3$N$_4$ interface as the signature of an inert OER catalytic surface, which is the bottleneck to prevent the continuous overall water splitting on the single-phase pristine g-C$_3$N$_4$ catalyst. Preventing the C=O accumulation via a simple surface fluorination strategy restored deserved overall water-splitting activity on fluorinated g-C$_3$N$_4$ catalysts with the H$_2$ evolution rate was order-of-magnitude improved compared to the pristine CN and continuous O$_2$ evolution upon both white light and AM1.5G simulated solar irradiation. Density functional theory (DFT) calculations were further employed to simulate the surface

[1]Institute of Green Chemistry and Chemical Technology, School of Chemistry and Chemical Engineering, Jiangsu University, 212013 Zhenjiang, People's Republic of China. [2]Department of Electrical Engineering and Computer Science, University of Michigan, Ann Arbor, MI 48109, USA. ✉e-mail: dgy5212004@163.com; dpzhou@umich.edu

fluorination-promoted OER at $H_2O$/CN interface and evaluate the impact of different intermediate OER configurations.

## Results and discussion

### In-situ observations of C=O accumulation at the $H_2O$/CN interface

The typical g-$C_3N_4$ catalyst (denoted CN) was prepared by sintering melamine powder at 550 °C in a muffle furnace according to a classic protocol from literature (for details of the preparation method, see Supplementary Information)[17]. We first used isotopic $^{16}O/^{18}O$-labeled $H_2O$ for in situ tracing of possible OER intermediate at the $H_2O$/CN interface during the continuous reaction by DRIFTS. $H_2O$ molecules were carried into the reaction chamber by $N_2$ flow until equilibrium. Setting the equilibrium condition as the blank background, positive or negative IR response signal directly reflects the gain or loss of intermediate species at the $H_2O$/catalyst interface. As shown in Fig. 1a, when the CN/$H_2O$ sample was used and irradiated in situ with a 420 nm LED lamp, a broad negative absorption band from 3700 $cm^{-1}$ to 3000 $cm^{-1}$ and a very weak negative peak at 1645 $cm^{-1}$ emerged from the background and increased in intensity with increasing irradiation time. The broad negative band was assigned to the stretching vibration of the O-H bond. In contrast, the weak negative peak at 1645 $cm^{-1}$ was from the bending vibration of H-O-H of $H_2O$ molecules, representing the loss of surface -OH species and $H_2O$ molecules during continuous OER[18,19]. The signal of O-H stretching vibration was much larger than the bending vibration signal of $H_2O$ molecules, suggesting that the OER that occurred at $H_2O$/CN interface was predominantly in the form of dissociated O-H. Identical features were observed when CN was replaced with $F_{0.1}$-CN or when $H_2O$ was replaced with $^{18}O$-labeled $H_2^{18}O$ (Fig. 1a–d) as the signature of OER at the $H_2O$/CN interface. More importantly, an increasing positive peak at 1725 $cm^{-1}$ ascribed to the C=O stretching vibration was observed with increasing irradiation time (Fig. 1a), indicating the collective formation of C=O species on the CN surface. When we replaced $H_2O$ with $^{18}O$-labeled $H_2^{18}O$ under otherwise identical conditions (Fig. 1b), the positive peak at 1725 $cm^{-1}$ and the newly generated peak at 1524 $cm^{-1}$ emerged in terms of the $^{16}O/^{18}O$ replacement effect according to Hooke's law[20], which confirms that the

O source of C=O was from $H_2O$ and further provides direct evidence for C=O accumulation during photocatalytic OER at $H_2O$/CN interface. Such a C=O formation can only occur with carbon sites on CN oxidized. To prevent the accumulation of C=O on CN, external atoms with vital bonding energy with C atoms are needed. We devised a surface fluorination strategy to occupy carbon sites on CN with $F^-$ ions through hydrothermal treatment. Prepared fluorinated CN samples (denoted F-CN) were labeled as $F_{0.01}$-CN-$F_1$-CN with different $F^-$ concentrations (0.01 mM–1 mM) of the precursor for fluorination (for details of the preparation method, see Supplementary Information). The surface fluorination did not severely change the morphology (Supplementary Fig. 1) and crystalline structure of CN (Supplementary Fig. 2) but formed a solid C–F interaction (Supplementary Fig. 3). When we replaced CN with $F_{0.1}$-CN (Fig. 1c), the positive C=O signal was no longer observed at the $H_2O$/$F_{0.1}$-CN interface. Further replacing the $H_2O$ with $^{18}O$-labeled $H_2^{18}O$ under otherwise identical conditions showed neither C=O nor C=$^{18}O$ diagnostic signals (Fig. 1d), which solidly confirms that the fluorination of CN prevents the carbon sites being oxidized into C=O intermediates. Similar phenomena were also observed when using the white light (Xe lamp) in replacement of the 420 nm irradiation (LED lamp, ≥300 nm) as the excitation light source (Supplementary Fig. 4) and with Pt cocatalyst loading before the measurements (Supplementary Fig. 5), which excludes the influence of the excitation wavelength and Pt loading on the interfacial reaction mechanism on CN and F-CN catalysts.

The collective formation of the C=O state by oxidizing carbon sites on CN was also directly observed by NAP-XPS. The NAP-XPS spectra were in situ collected in a vacuum chamber with 0.2 mbar $H_2O$ vapor pressure. A 300 W Xenon lamp as the white light source was placed outside the chamber to illuminate the sample via the quartz window. On the O1s spectra of the pristine CN sample, two major peaks at 530.1 eV and 531.3 eV were observed, corresponding to oxygen states of C-O and O-H species (Fig. 1e), respectively[21]. Under the white light illumination, a newly emerged contribution at 532.7 eV from C=O configuration was observed and increased in intensity with increasing irradiation time (Fig. 1e). Moreover, on the C1s spectra, peaks of C–C and N=C–N states on the pristine CN were gradually shifted towards

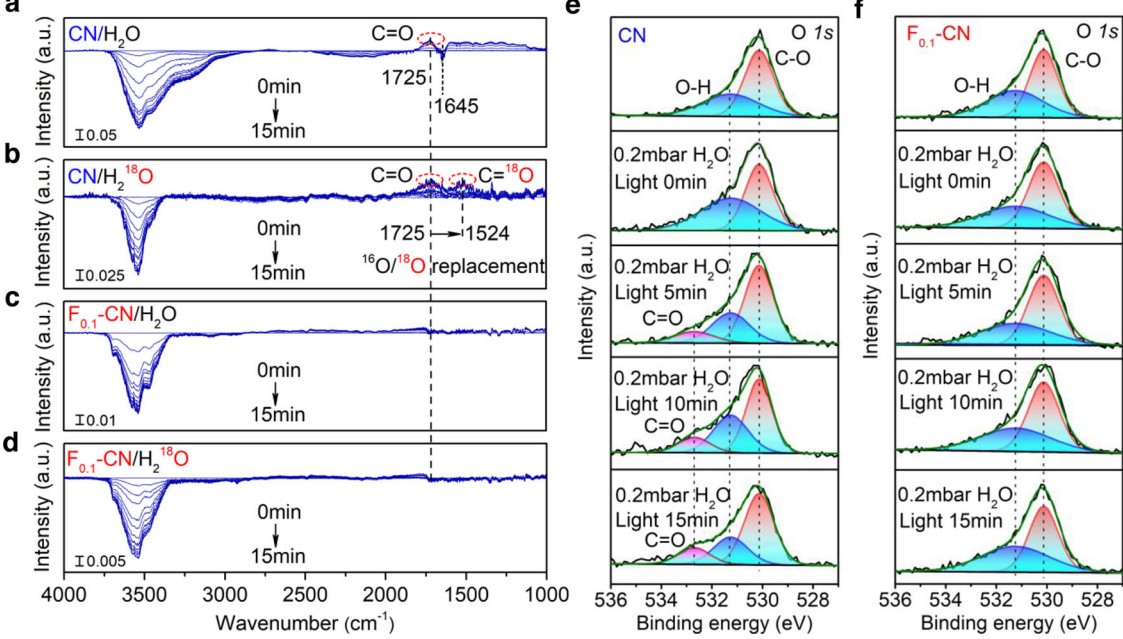

**Fig. 1 | In-situ observations of C=O accumulation at the $H_2O$/CN interface.** DRIFTS spectra in situ monitored at **a** CN/$H_2O$, **b** CN/$H_2^{18}O$, **c** $F_{0.1}$-CN/ $H_2O$, and **d** $F_{0.1}$-CN/ $H_2^{18}O$ interface under constant 420 nm (3 W, LED) irradiation in 15 min using pristine CN and the champion fluorinated F-CN ($F_{0.1}$-CN) catalysts. **e** In-situ NAP-XPS O1s spectra on pristine CN and **f** $F_{0.1}$-CN catalysts with 0.2 mbar $H_2O$ vapor pressure using a 300 W Xenon lamp as the white light source in 15 min.

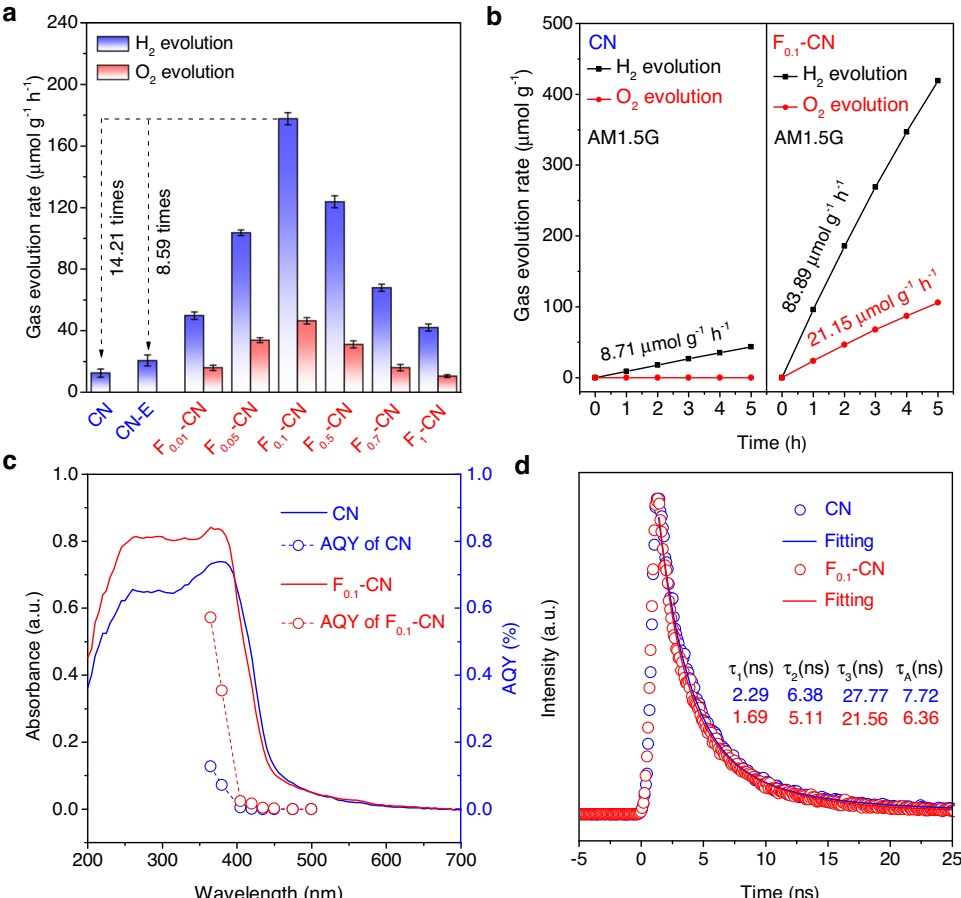

**Fig. 2 | Comparison of the overall water–splitting performances on CN/F-CN catalysts. a** Photocatalytic $H_2$ and $O_2$ productions from pure water on pristine CN, CN-E, and different F-CN catalysts under white light illumination. Error bars (in standard deviation) were obtained by statistically repeating identical experimental results three times. **b** Time profiles of photocatalytic $H_2$ and $O_2$ productions from pure water on pristine CN and $F_{0.1}$-CN under AM 1.5 G simulated solar irradiation. **c** Wavelength-dependent AQYs on pristine CN and $F_{0.1}$-CN, along with the corresponding UV–Vis DRS spectra. **d** Transient fluorescence emission decay at 465 nm on CN and $F_{0.1}$-CN catalysts with 375 nm excitation.

higher binding energy from 284.4 eV and 287.7 eV to 285 eV and 288.3 eV (Supplementary Fig. 6a, fitting parameters see Supplementary Table 1), respectively, under continuous white light illumination, corresponding to the formation of an oxidized carbon state on CN[22]. The NAP-XPS result is consistent with in situ DRIFTS observations (Fig. 1a, b), demonstrating that C=O intermediate state was indeed formed at the $H_2O$/CN interface during OER. After fluorination, although the solid C−F interaction can be recognized from the C1s peak shifting of the $F_{0.1}$-CN sample in comparison with CN (Supplementary Fig. 6a), little changes were found on both O1s (Fig. 1f) and C1s (Supplementary Fig. 6b) spectra of $F_{0.1}$-CN during continuous white light illumination with 0.2 mbar $H_2O$ vapor, which further demonstrates that C=O formation was vastly minimized on F-CN. In contrast, no changes were found in the N1s spectra on both CN and $F_{0.1}$-CN samples (Supplementary Fig. 6c, d). We further performed static DRIFTS (Supplementary Fig. 7) and XPS (Supplementary Fig. 8) measurements on CN, CN-E, and $F_{0.1}$-CN, before and after 5 h of water-splitting reaction. The C=O accumulation was not observed on CN and CN-E samples after the reaction, suggesting that C=O only formed during the OER as an intermediate species rather than be a stable surface group stoichiometrically produced in the reaction.

## Relationship between C=O bonding and overall water-splitting performances

We argue that the strong C=O bonding accumulation corresponds to the inert catalytic surface for OER on single-phased CN catalysts, an inherent bottleneck for the CN-based overall water splitting. If that is the case, preventing the intermediate C=O formation and activating the OER catalytic surface would endow CN catalysts with deserved overall water-splitting performances. Photocatalytic overall water-splitting experiments on CN and different F-CN samples were performed in pure water without any organic sacrificial reagents under both the white light (Fig. 2a) and AM1.5 G simulated solar irradiation (Fig. 2b). Under continuous white light irradiation (Xe lamp, >300 nm, spot center intensity 1000 mW $cm^{-2}$), the pristine CN catalyst only exhibited a mild $H_2$ evolution of 12.51 µmol $g^{-1}$ $h^{-1}$ without $O_2$ evolution. After hydrothermal treatment, CN exfoliated thin layer sample (denoted CN-E) showed a slightly higher $H_2$ evolution rate of 20.69 µmol $g^{-1}$ $h^{-1}$ due to the enlarged specific surface area of CN-E (62.12 $m^2$ $g^{-1}$) in comparison with CN (8.66 $m^2$ $g^{-1}$), but still, no $O_2$ evolution observed. The enlarged specific area of CN-E stems from the reduced layer thickness (Supplementary Fig. 9). The poor performance of CN and CN-E catalysts is consistent with literature reports[23,24], demonstrating that single-phased g-$C_3N_4$ catalyst does not possess the overall water-splitting ability. However, after the fluorination treatment, all F-CN catalysts exhibited both $H_2$ and $O_2$ evolution capabilities under identical experimental conditions, which varies with the fluorination degree. Notably, the $F_{0.1}$-CN catalyst showed an $H_2$ evolution rate of 177.79 µmol $g^{-1}$ $h^{-1}$, which is 14.21 and 8.59 times higher than those of the pristine CN and CN-E catalysts, respectively, and continuous $O_2$ evolution of 46.47 µmol $g^{-1}$ $h^{-1}$ (Fig. 2a). Although the specific surface area of $F_{0.1}$-CN (42.69 $m^2$ $g^{-1}$) is larger than that of the

pristine CN (8.66 $m^2 g^{-1}$) after hydrothermal exfoliation treatment due to the reduced thickness (Supplementary Fig. 9), it is still smaller than that of CN-E (62.12 $m^2 g^{-1}$) (Supplementary Fig. 10), yet $F_{0.1}$-CN exhibited order-of-magnitude-improved water-splitting efficiency (Fig. 2a, b). The larger loading of F ion will significantly reduce the performance after $F_{0.1}$-CN. We quantified the F ion content in F-CN samples using elemental analysis and ion chromatography. As summarized in Supplementary Table 2, from $F_{0.1}$-CN to $F_1$-CN, the F/C ratio was increased from 0.6 atom% to 11.4 atom%. To figure out the reason that F-CN catalyst with high F content (F/C > 0.6 atom%) further reduces the performance, we have carefully studied the changes in F-CN catalysts from low to high F content on the crystal structure, morphology, light absorption, electrochemical impedance, and hydrophilicity of the materials. XRD patterns (Supplementary Fig. 11a) and $N_2$ adsorption-desorption isotherms (Supplementary Fig. 11c) show that the crystalline structure and BET surface area of F-CN were not changed from low to high F content (from $F_{0.1}$ to $F_1$). As shown in Supplementary Fig.11b, the UV–Vis absorption of the F-CN catalyst was slightly blue-shifted with high F content, which may result in a slight drop in visible-light absorption. Furthermore, with the increase of F content, the electrochemical impedance (Supplementary Fig. 12a) increased, and the photocurrent decreased (Supplementary Fig. 12b), indicating that the inter-particle charge transfer ability decreased, which is one of the main reasons for the performance degradation of F-CN at high F content. We also systematically explored the relationship between F content and the hydrophilicity of F-CN materials and compared their suspension and sedimentation behaviors in water (Supplementary Fig. 13). The results show the hydrophilicity of F-CN significantly decreases (the water contact angle increases) with the rise of F content. The sedimentation time in water decreases with the growth of F content. The fast sedimentation behavior of the F-CN catalyst with high F content will render a lower light absorption efficiency than the well-suspended counterpart. Indeed, by using the in situ UV–Vis optical fiber spectroscopy to directly monitor the transmittance of white light (tungsten lamp, 5 W) through different suspensions (0.3 g/L), we observed a significantly increased transmittance of white light (30.03%–63.79%) with the increase of F-content ($F_{0.01}$–$F_1$) (Supplementary Fig. 14), which is another main reason for the performance decay of F-CN at high F content.

Moreover, under AM1.5 simulated solar irradiation (3 sun illumination), the $F_{0.1}$-CN catalyst still exhibited excellent overall water-splitting capacity with an $H_2$ evolution rate of 83.89 $\mu mol\, g^{-1} h^{-1}$, increasing by 9.63 times in comparison with the pristine CN catalyst (8.71 $\mu mol\, g^{-1} h^{-1}$), and continuous $O_2$ evolution rate of 21.15 $\mu mol\, g^{-1} h^{-1}$. Control experiments have been done to confirm that no $H_2/O_2$ productions were detected in the dark, no catalysts or without $H_2O$ for the $F_{0.1}$-CN catalyst (Supplementary Fig. 15). Isotopic-labeled experiments also confirmed that $H_2$ and $O_2$ were produced sorely from the photocatalytic water splitting rather than other effects, whereas $D_2$ and $^{18}O_2$ were detected as products of $D_2O$ and $^{18}O$-labeled $H_2^{18}O$ (Supplementary Fig. 16). Notably, $H_2/O_2$ production ratio on F-CN catalysts was higher than the stoichiometric ratio of 2:1 (Fig. 2a, b). The $H_2/O_2$ ratio was slightly increased with the increasing F-content in F-CN samples and still typically around 3-4 (Supplementary Fig. 17). The shortage of $O_2$ production on F-CN was possible due to the further reduction of $O_2$ into $H_2O_2$ since CN is very active for $O_2$ reduction[25,26]. We employed a $Ce^{4+}$ back titration method to quantify the $H_2O_2$ production during the reaction (Supplementary Fig. 18)[27]. The $H_2O_2$ production rate on the champion $F_{0.1}$-CN catalyst was determined to be 85.36 $\mu mol\, g^{-1} h^{-1}$ under white light, and 41.75 $\mu mol\, g^{-1} h^{-1}$ under AM1.5 G simulated solar irradiation, almost identical to the short of $O_2$ production. The reaction's solar to hydrogen (STH) efficiency was also calculated based on the $O_2/H_2O_2$ ratio and determined to be 0.00195% (details of the calculation method see Supplementary information). Furthermore, within 40 h, $F_{0.1}$-CN can still maintain more than 70% of efficiency on $H_2$ and $O_2$

production and continue to work (Supplementary Fig. 19). In contrast, CN and CN-E were quickly deactivated with less than 50% of initial efficiency on $H_2$ production within 8 h (Supplementary Fig. 20), indicating that the $H_2$ evolution on $F_{0.1}$-CN came from the continuous overall water splitting. In contrast, the mild $H_2$ evolution on CN and CN-E was possibly from unsustainable self-oxidation. The chemical state of F in $F_{0.1}$-CN after the reaction was also unchanged (Supplementary Fig. 21).

We compare the above photocatalytic performance results with our in situ DRIFTS (Fig. 1a–d) and in situ NAP-XPS (Fig. 1e, f) observations and reason that the accumulated intermediate C=O bonding directly corresponds to the deactivation of single-phased CN catalysts for overall water splitting. To further verify that the emerging overall water-splitting ability of F-CN stems from the prevention of C=O accumulation rather than other effects, we first compared the visible-light absorption of CN and F-CN catalysts. Figure 2c shows the wavelength dependence of apparent quantum yield (AQY) on the pristine CN and the champion $F_{0.1}$-CN catalysts, along with the ultraviolet–visible diffuse reflection spectra (UV–Vis DRS). As peak values, AQYs at 365 nm on both samples were determined to be 0.5718% ($F_{0.1}$-CN) and 0.1281% (CN). When the incident wavelength increased from 365 nm to 500 nm, the AQYs of both samples were sharply decreased (Supplementary Table 3 and Table 4), which coincides with literature reports on g-$C_3N_4$-based catalysts[28,29]. In the visible-light region, AQYs at 420 nm on both samples were determined to be 0.0164% ($F_{0.1}$-CN) and 0.0005% (CN). The much higher AQYs on the $F_{0.1}$-CN catalyst than that on the pristine CN catalyst further evince the effect of fluorination treatment. However, from UV–Vis DRS spectra, no discernible differences on the absorption edge between CN and $F_{0.1}$-CN were observed, indicating that the improved overall water-splitting performance of F-CN was not from the enhanced visible-light response. With higher F-content, the visible-light absorption on F-CN catalysts was slightly blue-shifted (Supplementary Fig. 11b), reducing the light absorption range.

We further tracked the transient fluorescence emission profile at 465 nm on $F_{0.1}$-CN and CN catalysts with the incident 375 nm irradiation. We found that the emission lifetime of the $F_{0.1}$-CN catalyst is not significantly extended in comparison with the pristine CN catalyst (Fig. 2d). The fitted emission decay profiles suggest a slightly shortened exciton lifetime on $F_{0.1}$-CN with a lifetime parameter reduced from $\tau_A = 7.72$ ns to $\tau_A = 6.36$ ns in comparison with the pristine CN, which denies the extended exciton lifetime as the significant effect of fluorination for enhanced overall water-splitting performances. Moreover, from the XPS valence band (VB) spectra near the Fermi level (Supplementary Fig. 22), CN and $F_{0.1}$-CN exhibited almost identical VB position at 1.88 eV, which denies the VB position as the major contributor for the order-of-magnitude performance improvement on F-CN.

Through the above characterizations, we ruled out that the morphology, crystalline structure, visible-light response, exciton lifetimes, and VB positions are the main factors affecting F-CN performance for overall water splitting. However, by using in situ DRIFTS (Fig. 1a–d) and NAP-XPS (Fig. 1e, f and Supplementary Fig. 4), we successfully identified the accumulation of C=O intermediate and their minimization on the F-CN surface, which is entirely consistent with the tendency of photocatalytic water-splitting performances. Furthermore, we performed the electrochemical OER experiment on CN, CN-E, and $F_{0.1}$-CN samples. The result shows that the F-modification indeed reduces the OER potential of $F_{0.1}$-CN in comparison with CN and CN-E (Supplementary Fig. 23). We further performed the HER (Supplementary Fig. 24) and OER half-reactions (Supplementary Fig. 25) with triethanolamine and $AgNO_3$ as the hole- and electron-acceptors, respectively. Results show that the OER production rate on F-CN was significantly enhanced, while the HER rate on F-CN was almost unchanged

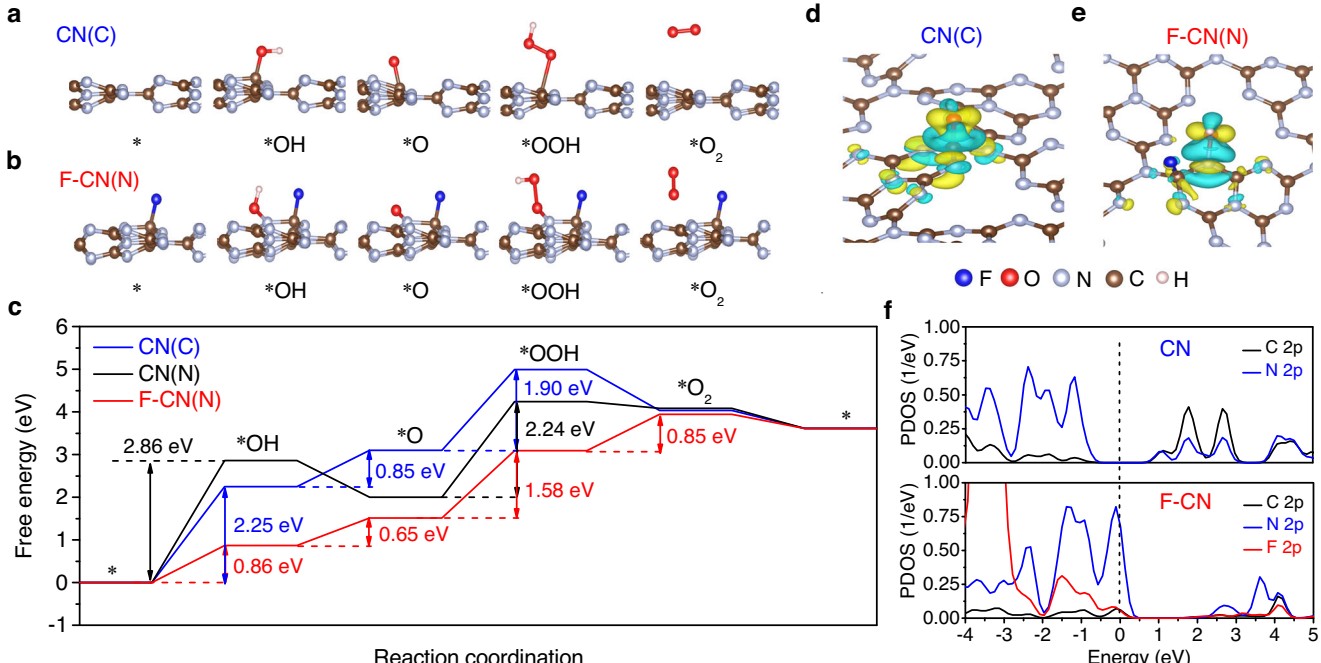

**Fig. 3 | Density functional theory calculations.** Water adsorption and activation were simulated on **a** the C site in pristine CN (denoted CN(C)) and **b** on the N site in F-CN (denoted F-CN(N)). **c** Free-energy profiles of OER on CN and F-CN at pH = 7 and $U = 0$ V vs. SHE (where * represents the intermediate state). CN(C) represents C reaction sites on the pristine CN; CN(N) represents N reaction sites on the pristine CN; F-CN (N) represents N reaction sites on F-CN (C reaction sites occupied entirely by F atoms). Charge density difference mappings between *OH intermediate and catalyst surface: **d** CN(C) and **e** F-CN(N). The blue and yellow isosurfaces stand for the negative and positive charges, respectively. The isosurface of charge density is set to 0.005 e Å$^{-3}$. **f** PDOS of 2p states of surface C, N, and F in CN and F-CN. The dashed line stands for the Fermi level.

compared to CN and CN-E catalysts, consistent with our hypothesis. Therefore, we conclude that the C=O bonding accumulation directly corresponds to an inert catalytic surface for overall water splitting on single-phased CN catalysts. By occupying the carbon site on the CN surface by fluorination, this bottleneck can be bypassed to achieve efficient visible-light-driven H$_2$ and O$_2$ productions from overall water splitting on single-phased F-CN catalysts.

## Density functional theory calculations

DFT calculations were further employed to investigate the effect of surface fluorination on the water decomposition reaction (i.e., OER) on the CN/F-CN surface (for details of the computational methods, see Supplementary Information). F-CN layer was formed by using one F atom to bond with the C atom in CN (Supplementary Fig. 26). Water adsorption and activation were simulated on both C sites (Fig. 3a) and N sites (Supplementary Fig. 27a) in pristine CN and N atoms adjacent to the C–F bond in F-CN as reactive sites (Fig. 3b). Calculated free-energy profiles show that OER on surface C site has a lower energy barrier (2.25 eV) than the surface N site (2.86 eV) in CN (Fig. 3c), indicating that C site is the predominant OER reactive sites in the pristine CN. Note that both transitions of *→*OH (2.25 eV) and *O → *OOH (1.90 eV) are high-barrier OER steps over CN (C). To significantly improve OER efficiency, lower energy barriers for both steps are necessary. After the F atom occupies the C site, the C site is saturated, which becomes inert to the reactant or intermediate. As a result, the neighboring two-coordinated N site is the sole catalytic center on F-CN. The obtained reaction pathway shows that the surface N site in F-CN owns much lower energy barriers for *→*OH (0.86 eV) and *O → *OOH (1.58 eV) than the surface C or N site in pristine CN (details of reaction energy differences see Supplementary Table 5), demonstrating that the F modification indeed can improve the OER activity on F-CN (Fig. 3c). Note that the F atom can bond with two kinds of C atoms (denoted $C_{3c}^1$ and $C_{3c}^2$) in F-CN (Supplementary Fig. 28)[30]. Thus, we calculated free-

energy profiles on N atoms with both F-occupied neighboring $C_{3c}^1$ and $C_{3c}^2$ atoms and obtained identical OER reaction energies (Supplementary Fig. 29). According to the calculated charge density difference mappings, the improved OER activity on F-CN is attributed to the more local charge distribution between CN surface and *OH intermediate, which significantly promotes the formation of *OH intermediate (Fig. 3e) and also effectively avoids the excessively strong C−O interaction (Fig. 3d) or weak N-O interaction (Supplementary Fig. 27b) in pristine CN. As a result, the F modification dramatically decreases the formation energy of rate-determining *OH. It should be noted that the F modification also significantly promotes the formation of *OOH, which is also a high-barrier reaction step in OER (Fig. 3c). This implies that the excessively stable *O intermediate in the form of C=O bond on pristine CN is challenging to be further converted into *OOH. During the OER over CN(C), both steps of *→*OH and *O → *OOH have high-energy barriers, which would lower the opportunity for the reaction. However, once the reaction starts with *OH formed, the subsequent formation of *O is accessible after a low-barrier step of *OH → *O. Thus, the accumulation of *O (in the form of C=O) can be observed since the opportunity of the further reaction step is relatively low with a high barrier (*O → *OOH). Once a small amount of *OOH is formed, it is quickly converted to O$_2$ with no −COOH accumulation observed. However, once the active N site emerged (adjacent to C−F), water-splitting reaction has a higher chance of occurring at the active N site than at regular C or N sites, which hinders the surface C=O accumulation. As a result, CN with a visual IR signal of C=O during the reaction owns a lower activity than F-CN, which coincides with our experimental observations. Furthermore, the PDOS calculation provides an electronic-scale insight into the improved OER activity on F-CN. The results show that F modification enables the N 2p states to move upward the Fermi level (Fig. 3f), which can be attributed to the transfer of partial electrons from the N site to the F site through the C site. The more positive N 2p states promise the N site with higher oxidizing

activity and more uncaptured orbits for bonding with *OH intermediate. Thus, the optimized bonding behavior between *OH and F-CN surface contributes to an improved OER activity. Hence, the N site in F-CN is the main OER center. DFT calculations with varying F contents (with surface F coverage in the range of 1-8 atom%) were also conducted (the optimal $F_{0.1}$-CN catalyst has a surface F coverage of ~4 atom%) (Supplementary Table 1). As shown in Supplementary Fig. 21a, with surface F coverage increased from 1 atom% to 8 atom%, the barrier of OER on the adjacent N site (Supplementary Fig. 30a), and the energy level of N 2p state (Supplementary Fig. 30b) was almost unchanged, indicating that the adjacent N site of the C−F structure is only affected by the adjacent C−F hybridization rather than the concentration of F. The F coverage only affects the number of active N sites.

The unexpected C=O accumulation by oxidizing surface carbon atoms during the OER on single-phased CN catalysts is a previously unrecognized event. As a signature of the inert catalytic surface for OER, such a strong bonding of intermediate C=O directly corresponds to the deactivation of single-phased CN catalysts, which actively acts as the bottleneck of overall water splitting on CN-based catalysts. Our present study identifies the unrevealed cause responsible for the water-splitting deactivation on single-phased CN catalysts. To bypass this bottleneck, a simple and robust surface fluorination treatment to suppress C=O accumulation by forming C−F and lower the OER barriers by activating adjacent N reactive sites was devised, which significantly restores the deserved overall water-splitting ability under visible light on resulting F-CN catalysts.

## Methods
### Materials
Melamine ($C_3H_6N_6$), sodium fluoride (NaF), perchloric acid ($HClO_4$), nitric acid ($HNO_3$), and ethanol ($C_2H_5OH$) were purchased from Sinopharm Chemical Reagent Co. Ltd. Chloroplatinic acid hexahydrate ($H_2PtCl_6 \cdot 6H_2O$), deuterium oxide ($D_2O$, 99 atom % D) and heavy oxygen water ($H_2{}^{18}O$, 97 atom % $^{18}O$) were obtained from Sigma-Aldrich Company Ltd. All reagents used in the synthesis were analytical grade and used without further purified. Deionized water, with a resistivity of 18 MΩcm, was used throughout the experiments.

### Preparation of pristine CN
Typically, CN was synthesized by calcining melamine powder (10 g) at 550 °C for 4 h with a ramping rate of 5 °C min$^{-1}$ in static air in a muffle furnace. After naturally cooling down to room temperature, the resulting product was dissolved into 0.5 M $HNO_3$ solution, bathing for 4 h at 80 °C. Then, light yellow agglomerates were gathered, cross-washed with deionized water and ethanol several times, and dried at 60 °C in a vacuum oven for further use.

### Preparation of F-CN
$F_X$-CN samples were prepared by a one-step hydrothermal treatment of pristine CN. In a typical procedure, the pristine CN (1 g) was mixed in aqueous solutions containing different concentrations of NaF (0.01-1 mM) with stirring for 30 min, and the pH was adjusted to 3.5 by $HClO_4$. Then, the resulting solution was transferred to a sealed Teflon container at 180 °C for 12 h. The final sample was labeled as $F_X$-CN ($x$ represents the NaF concentration employed). Exfoliated CN (CN-E) sample was prepared under otherwise identical conditions without NaF.

### Characterizations
In-situ diffuse reflection infrared Fourier transform spectroscopy (DRIFTS) experiments were performed on a Thermo Nicolet iS10 spectrometer equipped with a mercury cadmium telluride (MCT) detector. The near-ambient pressure X-ray photoelectron spectroscopy (NAP-XPS) experiments were conducted on a laboratory-based SPECS near-ambient pressure XPS system. The XRD patterns were examined on a Shimazu-6100 powder X-ray diffractometer using Cu Kα radiation at a scan rate of 7° min$^{-1}$. Electron Microscope (TEM) analyses were performed on an H-7800 microscope with an acceleration voltage of 120 kV. The ultraviolet−visible diffuse reflectance spectroscopy (UV−Vis DRS) was recorded on a Shimadzu UV-3600 spectrometer with $BaSO_4$ as a reference. The isotopically labeled experiments were carried out on a gas chromatography-mass spectrometry (GC-MS, Agilent 7890B-5977B). Transient fluorescence decay spectra were characterized on an Edinburgh FLS1000 fluorescence spectrometer. $N_2$ adsorption-desorption isotherms were obtained on a BEL SORP mini (Microtrac BEL, Japan).

### Photocatalytic water-splitting experiments
Photocatalytic overall water-splitting reactions were carried out in a top-irradiation Pyrex reactor connected to a glass-sealed gas system with a 300 W Xenon lamp (≥300 nm, 1000 mW cm$^{-2}$) as the white light source. 30 mg catalyst was dispersed in 100 ml of pure water, and 3 wt% Pt as co-catalysts was loaded via in situ photo-deposition using $H_2PtCl_6 \cdot 6H_2O$ without any sacrificial agents recirculating cooling water at 6 °C. The gas productions were monitored by gas chromatography assembled with a thermal conduction detector (TCD, 5 A molecular sieve columns) using Ar as carrier gas. All reactions were carried out at 279 K in a low vacuum.

### In situ DRIFTS experiments
In situ diffuse reflection infrared Fourier transform spectroscopy (DRIFTS) experiments were implemented on a Thermo Scientific Nicolet iS10 spectrometer equipped with a mercury cadmium telluride (MCT) detector. In a typical procedure, the catalyst was housed in a sample groove inside a sealed reaction cell with two ZnSe windows and a quartz window, as well as $H_2O$ was carried into the reaction cell by $N_2$ flow until equilibrium. IR spectra were recorded during the in situ photoreactions with a 420 nm LED lamp (3 W) as a light source through the quartz window. Before isotopic experiments, pristine CN and F-CN samples were heated at 423 k for 30 min under flowing $N_2$ to remove the remaining water, which was used for the IR test to exclude $H_2O$/$H_2{}^{18}O$ exchange. Samples were then cooled down to room temperature before IR measurements. All in situ DRIFTS measurements were conducted at 25ºC under ambient pressure conditions.

### In situ NAP-XPS experiments
The near-ambient pressure X-ray photoelectron spectroscopy (NAP-XPS) experiments were conducted on a laboratory-based SPECS near-ambient pressure XPS system (base pressure $<5 \times 10^{-10}$ mbar) with a monochromatic Al Kα X-ray source, equipped with an in situ reaction chamber that permits specific gas to flow the reaction chamber during data acquisition. The catalyst, anchored at a Ta-based lame of the groove inside the reaction chamber, and $H_2O$ was carried into the samples via a variable leak valve connected to a glass bulb containing MilliQ water (18 MΩ cm$^{-1}$), and the water was freeze-degassed several times using liquid nitrogen before the reaction. The in situ reaction chamber collected the NAP-XPS spectra with 0.2 mbar $H_2O$ vapor pressure using a 300 W Xenon lamp as the white light source in 15 min through the quartz window. All NAP-XPS spectra were calibrated to the Ta $4d_{5/2}$ peak at 230 eV.

### H2O2 quantification
The $H_2O_2$ production was quantified by the $Ce^{4+}$ back titration method based on the mechanism that a yellow solution of $Ce^{4+}$ can be reduced by $H_2O_2$ to colorless $Ce^{3+}$ (Eq. (1)). The absorption peak of the concentration of $Ce^{4+}$ before and after the reaction used for the measurement was 316 nm via the UV−Vis spectrophotometer.

$$2Ce^{4+} + H_2O_2 \rightarrow 2Ce^{3+} + 2H^+ + O_2 \tag{1}$$

Therefore, the concentration of $H_2O_2$ can be obtained by Eq. (2):

$$M = \frac{1}{2} \times M\text{Ce}^{4+} \tag{2}$$

where $M\text{Ce}^{4+}$ is the mole of consumed $\text{Ce}^{4+}$.

In a typical procedure, the yellow transparent $Ce(SO_4)_2$ solution (1 mM) was confected by dissolving 83 mg $Ce(SO_4)_2$ in 250 mL 0.5 M sulfuric acid solution. To obtain the calibration curve, $H_2O_2$ with known concentration was added to $Ce(SO_4)_2$ solution and measured by a UV–Vis spectrophotometer. Thus, the $H_2O_2$ concentrations of the samples could be realized by the linear relationship between the signal intensity and $\text{Ce}^{4+}$ concentration.

## STH efficiency calculation

$$\text{STH(\%)} = \frac{\text{Energy of generated } H_2}{\text{Solar energy irradiating the solution}} = \frac{R_{H_2} \times \Delta G_r}{P_{sun} \times S} \tag{3}$$

Here, $R_{H_2}$ is the rate of hydrogen evolution under the AM1.5G irradiation. $\Delta G_r$ is a Gibbs energy of water-splitting reaction, $P_{sun}$ is the optical power density of standard spectrum with AM1.5 G. $S$ is the irradiation area. $\Delta G_r$ can be obtained by follow equations.

$$2H_2O(l) \rightarrow 2H_2(g) + O_2(g)\ \Delta G_1 = 237.1 \text{KJ} \cdot \text{mol}^{-1}$$

$$H_2O(l) + \frac{1}{2}O_2(g) \rightarrow H_2O_2(l)\ \Delta G_2 = 116.7 \text{KJ} \cdot \text{mol}^{-1}$$

$$2H_2O(l) \rightarrow H_2(g) + H_2O_2(l)\ \Delta G_3 = 235.3 \text{KJ} \cdot \text{mol}^{-1}$$

So,

$$
\begin{aligned}
\text{STH(\%)} &= \frac{R_{H_2} \times \Delta G_r}{P_{sun} \times S} \\
&= \frac{R_{H_2}^1 \times \Delta G_1}{P_{sun} \times S} \times 100\% + \frac{R_{H_2}^3 \times \Delta G_3}{P_{sun} \times S} \times 100\% \\
&= \frac{42.3 \times 0.03 \times 10^{-6} \times 237.1 \times 10^3}{3600 \times 0.3 \times 28.26} \times 100\% \\
&\quad + \frac{41.75 \times 0.03 \times 10^{-6} \times 235.3 \times 10^3}{3600 \times 0.3 \times 28.26} \times 100\% = 0.00195\%
\end{aligned}
$$

## Computational methods

The free-energy profiles of OER on CN and F-CN were investigated by the Vienna Ab-initio Simulation Package (VASP) with the revised Perdew–Burke–Ernzerh functional of (RPBE) of the generalized gradient approximation (GGA). PAW pseudo-potential describes the interaction between ionic core and valence electrons. A $2 \times 3$ supercell was used to simulate the CN layer. One F atom was bonded with the C atom in CN, forming the F-CN layer. The energy cutoff of the plane-wave basis of 400 eV and the energy convergence threshold of $1.0 \times 10^{-5}$ eV were used in the geometry optimization at the gamma point. After geometry optimization, the projected density of states (PDOS) and the charge density difference mappings between the intermediate and photocatalyst surface were calculated with an energy convergence threshold of $1.0 \times 10^{-5}$ eV and the energy cutoff of the plane-wave basis of 400 eV at gamma point. The calculations of Gibbs free-energy changes ($\Delta G$) of all reaction steps adopted the reported standard hydrogen electrode (SHE) model[31], which was obtained from the following formula:

$$G = E + \text{ZPE} - TS \tag{4}$$

where $E$, ZPE, and $S$ are the electronic free-energy, zero-point energy, and model entropy at $T = 298.15$ K, respectively.

## Data availability
The data supporting the findings of this study are available within the article and its Supplementary Information files. All other relevant source data are available from the corresponding author upon request following the data management specifications of Jiangsu University and University of Michigan. Source data are provided with this paper.

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

## Acknowledgements

We are grateful for the technical support from Dr. Yifan Li and Prof. Yi Cui of Nano-X from Suzhou Institute of Nano-Tech and Nano-Bionics, Chinese Academy of Sciences (SINANO). We gratefully acknowledge the financial support of the National Natural Science Foundation of China (Grant No. 21776117 (P. Huo) and 21806060 (Y. Yan)).

## Author contributions

J.W. and Y.Y. designed the whole experiment. J.W., Z.L., X.L., E.J., and S.Z. conducted most experiments. J.W. and Y.Y. wrote the paper. P.Z. contributed to the DFT calculation. P.H., P.Z., and Y.S.Y. contributed to the data analysis of the paper quality through discussions.

## Competing interests

The authors declare no competing interests.
