## [Peer Review File · Nature Communications]

Breaking Through Water-Splitting Bottlenecks Over Carbon Nitride with FluorinationReviewers' Comments:

Reviewer #1:

Remarks to the Author:

In this manuscript, the authors presented an interesting work on revealing the inhibiting effect on carbon nitride performance in photocatalytic water splitting to H₂ and O₂. The authors employed in-situ FTIR and XPS with isotopic detection for the identification of C=O formation in the process, which was suggested to play an important role for catalytic reaction. This work is new and can be published after revision. Some suggestions are listed below.

1. It is believed that C=O formation on the surface of carbon nitride. Thus, it is suggested that the authors check the surface species on the catalysts (CN, CN-E and F-CN) before and after reaction to see the difference and discuss the effect of C=O loading on their catalytic performance.
2. What is the mechanism of C=O involving in the catalytic reaction to reduce the performance?
3. page 3, line 3 of the first paragraph. "F ion would slightly decrease the performance" is not correct. From Fig.1a, larger loading of F ion will significantly reduce the performance after F0.1.
4. page 3, line 12 of the first paragraph. "was less than stoichiometric ratio of 2:1" is not correct. Fig.1a shows that H₂:O₂ is much larger than 2:1.
5. The authors suggested that high loading of F in carbon nitride results in low activity because of hydrophobicity, which is not convinced and may not be the intrinsic nature. The authors should consider other characteristics of the catalyst and checked the properties. If possible, the authors should also carry out DFT calculations with more F in the structure.

Reviewer #2:

Remarks to the Author:

The authors report that the C=O formation during the OER is the bottleneck of photocatalytic overall water splitting on CN, while this C=O bonding could be hindered through F modification. In-situ XPS and DRIFTS characterizations were also carried out. This assumption is innovative, but there are still several fundamental concerns need to be addressed.

1. In Fig. 3c, the rate-determining step of O₂ evolution over CN(C) is the formation of *OH rather than the step from O* to *OOH. So, why the formation of C=O is the bottleneck of overall water splitting on CN-based catalysts? More illustration or an additional attention on the effect of C=O should be given out.
2. The authors proposed that the N site is the main OER center in F-CN. While N 1s XPS spectra indicates no obvious shift under continuous white light illumination. Why an oxidized N state did not form like C in CN?
3. Since N is original present in F-CN and F modification is proposed to induce the change of reactive sites from C atoms in CN to N atoms adjacent to the C-F bond in F-CN, a fundamental discussion on the mechanism should be given out. The change of energy barrier from theoretical simulation is more like phenomenon other than underlying comprehension.
4. Why the upward of Fermi level can enhance the stability of formed *OH on the N site in F-CN? Characterizations should be carried to support the that "Fermi level moved upward".
5. Fig. 2c shows that the main activity difference of CN and F-CN lies in UV part. This conflicts with the pathway change proposed by the authors, because it is hardly to image that O₂ evolution proceeds in different ways under UV and visible light on F-CN.
6. It seems that H₂/O₂ production ratio on F-CN catalysts was larger than the stoichiometric ratio even taking account of the formation of H₂O₂. Why?

7. As we all know, there are two types of C atoms in CN. The authors are suggested to clarify which kind of C is modified by the F atom.

8. Can the author give more direct and reliable evidence to prove the formation of C-F bond?

9. Crystal defects including N vacancy (VN) in CN have been well studied, and their effects in photocatalysis have also been demonstrated. So, the authors are suggested to evaluate the possible effect of defects in this system.

Reviewer #3:

Remarks to the Author:

This is a very interesting study aiming at exploring the critical issue of oxygen evolution reaction on graphitic carbon nitride. The authors claimed that the formation of C=O bonding during continuous photocatalytic overall water splitting is the origin of carbon nitride failing to produce oxygen without the addition of small molecule organics. DRIFTA and NAP-XPS were used to support the claims, while fluorinated CN catalyst was designed to show that avoiding the formation of C=O can lead to the evolution of oxygen on carbon nitride without a sacrificing reagent. DFT calculations were also used to provide theoretical evidence. In general, the paper was well prepared, and can be considered for publication after some consideration of following issues.

(1) The fluorinated CN catalyst, i.e., F-CN, is a critical sample for the study, however the characterization is not sufficient. Also, AFM, photocurrent, EIS should be performed for all the samples in study.

(2) The chemical state of fluorine on the sample should be carefully explained. The XPS F1s spectra (Fig S3(b)) only show very weak signals with a high s/n ratio. The other concern is how the fluorine evolves after the reaction, given the fact that oxygen and hydrogen peroxide would appear.

(3) The stability of gases production, and the properties of the photocatalysts should be checked.

(4) If just aiming at the production of oxygen, other methods like loading cocatalyst on carbon nitride can also play the same role. What is the necessity of fluorination? How is this compared to other methods? How can this approach inspire or help other scientific challenges in materials chemical or science?

Reviewer #4:

Remarks to the Author:

This manuscript by Wu et al. reports a study on the reasons leading to the overall water splitting inactivation on single-phase CN catalysts. The authors claimed that the formation of inert C=O intermediates by oxidizing surface carbon atoms during the OER is the bottleneck for overall water splitting on g-C₃N₄ catalyst. After analyzing isotopic-labelled in-situ diffuse reflection infrared Fourier transform spectroscopy and in-situ near-ambient pressure X-ray photoelectron spectroscopy, the formation of C=O intermediates is prevented via a simple surface fluorination strategy, and the F0.1-CN catalyst exhibits higher overall water splitting activity compared to the pristine CN. However, many mechanisms and modification strategies for g-C₃N₄ to improve the photocatalytic overall water splitting activity have been reported in previous literatures (Nat. Energy 6, 388(2021); Adv. Mater. 33, 2007479(2021); Chin. Chem. Lett. 32, 13(2021)). The novelty and significance of this work should be highlighted because the catalytic activity of F0.1-CN catalyst is general compared to the catalysts reported so far. And some main conclusions are not well supported by the present experimental/theoretical results. I consider that the manuscript cannot meet the high standards of Nat. Commun., so I cannot recommend the publication of this work on the Nat. Commun.

1. Up to now, the nonmetallic-elements modification of g-C₃N₄-based catalysts and the corresponding

catalytic mechanisms have been widely studied (Nat. Energy 6, 388(2021); Small 17, 2006851(2021); Adv. Mater. 33, 2007479(2021); Chin. Chem. Lett. 32, 13(2021)). Therefore, where are the advance and major breakthrough of the views mentioned by the author in relation to the reported results?

2. The F0.1-CN catalyst exhibits poor overall water splitting activity (H_2 evolution rate of $174.77 \mu\text{mol g}^{-1} \text{h}^{-1}$) compared to the catalysts reported so far (H_2 evolution rate $> 500 \mu\text{mol g}^{-1} \text{h}^{-1}$, $\lambda > 420\text{nm}$; Small 17, 2006851(2021); Adv. Mater. 32, 1907296(2020)). Using isotopic-labelled in-situ diffuse reflection infrared Fourier transform spectroscopy and in-situ near-ambient pressure X-ray photoelectron spectroscopy to study mechanism is of little significance.

3. According to the signal-to-noise ratio of F1s spectra, the content of F should be very low for the F0.1-CN catalyst ($< 0.1\%$). The authors believed that the improved OER activity is due to the C-F bond formation optimizing the OER pathway on adjacent N atoms. However, the number of F-modified N sites should be minimal. The coordination environment of most carbon sites in F0.1-CN is similar to that in pristine CN. Therefore, it is not convincing that F-CN does not generate C=O intermediates during the reaction. The authors should provide more experimental and theoretical results to explain this issue in detail.

4. In the in-situ NAP-XPS observations of N1s spectra, the peak area ratios of NH_x , NC_3 , $C-N=C$ change significantly for the pristine CN and F0.1-CN catalysts. This phenomenon should be explained in detail. In addition, the authors claimed that the N site in F-CN is the main OER active center. However, during continuous white light illumination, there is no peak positions change or new peaks appear on the N1s spectra for the F0.1-CN catalyst, which is unreasonable.

5. In the in-situ DRIFTS analysis, the authors claimed that the O source of C=O is from H_2O . However, an obvious peak of the $C=^{16}O$ stretching vibration is still observed at 1725 cm^{-1} when $H_2^{16}O$ is replaced by ^{18}O -labelled $H_2^{18}O$. Furthermore, why does the bending vibration of H-O-H at 1645 cm^{-1} disappear in Fig. 1b-1d? More detailed analysis should be provided.

6. In the DFT calculations, the authors only compared the energy barrier of the surface N sites of F-CN and the surface C and N sites of pristine CN. Why did the authors not consider the C atom of the C-F bond in F-CN as the reaction site?

7. The authors should improve the accuracy of the expression. For example, what is the wavelength of white light?

Reviewer #5:

Remarks to the Author:

In this manuscript, the authors observed a collective C=O bonding during continuous photocatalytic overall water splitting on g-C₃N₄ catalyst by in-situ DRIFTS and NAP-XPS, and confirmed that the inert C=O bond directly hinders further OER steps, resulting in negligible O generation on CN. The F0.1-CN catalyst prepared based on this finding exhibited excellent overall water splitting activity with the order-of-magnitude improved H_2 evolution rate compared to the pristine CN catalyst. This manuscript was interesting. However, I feel that there are still a number of uncertainties and obvious problems. The analysis of the data is not accurate enough. After careful evaluation, I think that this work does not meet the standard of Nature Communications.

Other comments:

1. In-situ NAP-XPS is the main method for the authors to prove the formation and disappearance of C=O bonds. This method is very interesting, but I think there are some issues with data analysis on NAP-XPS. For example, in the XPS spectra of CN under different illumination times, the fitted peak

width of the O-H peak changed significantly. Actually, the total spectra at 0 min and 5 min of CN are close, but the C=O peak is formed due to the change in the peak width of the O-H peak during fitting. Authors should fit the XPS data under the same O-H peak width to determine the presence of C=O peak.

2. The consumption of e⁻ and h⁺ during photocatalytic overall water splitting should be equal.

However, in this work, the e⁻ consumed by H₂ production is not equal to the h⁺ consumed by O₂ and H₂O₂ production. It is suggested that the authors calculate the amount of reduction and oxidation products in the photocatalytic water splitting process and analyze why the consumption ratio of e⁻ and h⁺ is not 1:1.

3. The authors should point out the test conditions of Supplementary Figure 9, white light or AM1.5G?

4. The authors aim to illustrate the effect of F doping on enhanced OER performance, then supplemental separate OER tests may be more helpful in illustrating the authors' hypothesis.

5. The nanosheets of CN in Supplementary Figure 1 appear thinner than F-CN, but its BET surface is much smaller than that of F-CN. What is the reason for the increase in the specific surface area of F-CN?

6. It is difficult to illustrate the existence of F in Supplementary Figure 3. It is recommended that the authors re-test XPS and consider changing the sample preparation method to improve the accuracy.

Reply to reviewers' comments

To Reviewer 1:

In this manuscript, the authors presented an interesting work on revealing the inhibiting effect on carbon nitride performance in photocatalytic water splitting to H₂ and O₂. The authors employed in-situ FTIR and XPS with isotopic detection for the identification of C=O formation in the process, which was suggested to play an important role for catalytic reaction. This work is new and can be published after revision. Some suggestions are listed below.

Response: Thanks for your great efforts in reviewing our manuscript. We appreciate your valuable comments and suggestions.

Q1: It is believed that C=O formation on the surface of carbon nitride. Thus, it is suggested that the authors check the surface species on the catalysts (CN, CN-E, and F-CN) before and after reaction to see the difference and discuss the effect of C=O loading on their catalytic performance.

Response: Thanks for your valuable comment. We have carefully performed DRIFTS and XPS measurements on CN, CN-E, and F_{0.1}-CN, before and after 5 hours of reaction. As shown in Fig. R1 and Fig. R2, IR and XPS spectra of samples before and after the reaction remain unchanged. The C=O accumulation was not observed on CN and CN-E samples after the reaction, which further suggests that C=O only formed during the OER as an intermediate species (only observable during the reaction) rather than be a stable surface group stoichiometrically produced in the reaction. We have added the additional data and discussion in the revised manuscript. Please see the **Line 29 of Page 5**.

Fig. R1. The comparison of DRIFTS spectra on (a) CN, (b) CN-E, and (c) F_{0.1}-CN samples before and after 5 hrs of reaction.

Fig. R2. The comparison of XPS spectra on (a) CN (C1s and N1s), (b) CN-E (C1s and N1s), and (c) F_{0.1}-CN (C1s, N1s, and F1s) samples before and after 5 hrs of reaction.

Q2: What is the mechanism of C=O involving in the catalytic reaction to reduce the performance?

Response: Thanks for your valuable comment. According to our *in-situ* DRIFTS (Fig. 1a-1b) and NAP-XPS (Fig. 1e) observations, C=O species collectively formed during continuous photocatalytic reaction on CN with pure water by oxidizing carbon sites. During this process, C=O is generated as the reaction intermediate of OER at the C site (*O). The subsequent OER reaction requires the dissociation of the C=O double bond to combine with the second water molecule (or OH⁻ ion), forming the *OOH intermediate. Since the inert C=O double bond is thermodynamically difficult to dissociate, as a reaction intermediate, the C=O formation directly lowers the opportunity for OER.

Q3: page 3, line 3 of the first paragraph. "F ion would slightly decrease the performance" is not correct. From Fig. 1a, larger loading of F ion will significantly reduce the performance after F_{0.1}.

Response: Thanks for your valuable comment. The corresponding description error was corrected. Please see the **Line 4 of Page 7**.

Q4: page 3, line 12 of the first paragraph. "was less than stoichiometric ratio of 2:1" is not correct. Fig. 1a shows that H₂:O₂ is much larger than 2:1.

Response: Thanks for your valuable comment. The corresponding description error was corrected. Please see **Line 30 of**

Q5: The authors suggested that high loading of F in carbon nitride results in low activity because of hydrophobicity, which is not convinced and may not be the intrinsic nature. The authors should consider other characteristics of the catalyst and check the properties. If possible, the authors should also carry out DFT calculations with more F in the structure.

Response: Thanks for your valuable comment. We have carefully studied the changes in F-CN catalysts from low to high F content on the crystal structure, morphology, light absorption, electrochemical impedance, and hydrophilicity of the materials to further explore the reasons for the performance decay of samples with high F content. First, from XRD patterns (**Fig. R3a**), the crystalline structure of F-CN was not changed from low to high F content. As shown in **Fig. R3b**, the UV-vis absorption of the F-CN catalyst was slightly blue-shifted with high F content, which may result in a slight drop in visible light absorption and lower performance. The BET surface area of F-CN samples was also decreased with the increasing F content (**Fig. R3c**). However, the F₁-CN sample has a higher BET surface area (48.68 m²g⁻¹) than that of the optimized F_{0.1}-CN (42.69 m²g⁻¹) sample, indicating that the changed BET surface area is not the major effect on the performance decay. Furthermore, with the increase of F content, we found that the electrochemical impedance (**Fig. R4a**) increased and the photocurrent decreased (**Fig. R4b**), indicating that the inter-particle charge transfer ability decreased, which is one of the main reasons for the performance degradation of F-CN at high F content.

We also systematically explored the relationship between F content and the hydrophilicity of F-CN materials and compared their suspension and sedimentation behaviors in water (**Fig. R5**). The results show that the hydrophilicity of F-CN greatly decreases (the water contact angle increases) (**Fig. R5b**) with the increase of F content, and the sedimentation time in water decreases with the increase of F content (**Fig. R5a**). At high F content, F-CN is easier to agglomerate and settle down faster in water, which is another main reason for the performance decay of F-CN at high F content.

DFT calculations with varying F contents (with surface F coverage in the range of 1~8 atom%) were also conducted (The optimal F_{0.1}-CN catalyst has a surface F coverage of ~2 atom%). As shown in **Fig. R6**, with surface F coverage increased from 1 atom% to 8 atom%, the barrier of OER on the adjacent N site (**Fig. R6a**) and the energy level of N 2p state (**Fig. R6b**) were almost unchanged, indicating that the adjacent N site of the C-F structure is only affected by the adjacent C-F hybridization rather than the concentration of F. The F coverage only affects the number of active N sites.

In Summary, reduced inter-particle charge transfer capacity and hydrophilicity (resulting in difficult dispersion of the catalyst in water) are responsible for the performance decay of F-CN at high F content. The thermodynamical barrier and valence band on F-CN for OER are not affected by the F content. The above discussion, additional experimental data, and calculation results were added to the revised manuscript. Please see **Line 8 of Page 7** and **Line 21 of Page 10**.

Fig. R3. (a) XRD patterns, (b) UV-vis DRS spectra, and (c) N₂ adsorption-desorption isotherms of CN, CN-E and F-CN samples with different F contents. Inset of (c) shows the BET surface area of different samples.

Fig. R4. (a) EIS and (b) photocurrent curves of CN-E, F_{0.1}-CN, and F₁-CN sample.

Fig. R5. (a) The comparison of continuous sedimentation of CN-E and F-CN samples with different F content, and water surface contact angle on (b) CN-E, (c) F_{0.01}-CN, (d) F_{0.05}-CN, (e) F_{0.1}-CN, and (f) F₁-CN.

Fig. R6. (a) Free energy profiles of OER on F-CN (N) with F surface coverage of 1 atom% to 8 atom% (denoted F1~F8). F-CN (N) represents N reaction sites on F-CN (C reaction sites occupied entirely by F atoms). (b) PDOS of N 2p states in F-CN with F surface coverage of 1 atom% to 8 atom% (denoted F1~F8). The dashed line stands for Fermi level.

The authors report that the C=O formation during the OER is the bottleneck of photocatalytic overall water splitting on CN, while this C=O bonding could be hindered through F modification. In-situ XPS and DRIFTS characterizations were also carried out. This assumption is innovative, but there are still several fundamental concerns need to be addressed.

Response: Thanks for your great efforts in reviewing our manuscript. We appreciate your valuable comments and suggestions.

Q1: In Fig. 3c, the rate-determining step of O₂ evolution over CN(C) is the formation of *OH rather than the step from O* to *OOH. So, why the formation of C=O is the bottleneck of overall water splitting on CN-based catalysts? More illustration or an additional attention on the effect of C=O should be given out.

Response: Thanks for your valuable comment. As shown in **Fig. 3c**, during the OER over CN(C), the two steps of *→*OH and *O→*OOH have high-energy barriers, which would definitely lower the opportunity for the reaction. However, once the reaction starts with *OH formed, the subsequent formation of *O is easy after a low-barrier step of *OH→*O. Thus, the accumulation of *O (in the form of C=O) can be observed since the opportunity of the further reaction step is rather low with a high barrier (*O→*OOH), and once a small amount of *OOH is formed, it is quickly converted to O₂ with no -COOH accumulation observed, which is consistent with our experimental observations. We have added this part of the discussion to the revised manuscript. Please see the **Line 6 of Page 10**.

Q2: The authors proposed that the N site is the main OER center in F-CN. While N 1s XPS spectra indicates no obvious shift under continuous white light illumination. Why an oxidized N state did not form like C in CN?

Response: Thanks for your valuable comment. The unchanged N 1s state in F-CN indicates that the charge transfer at the N site was very fast with no obvious accumulation of intermediate states. This result further suggests that the N site as the main OER center facilitates the interfacial charge transfer after the surface F-modification.

Q3: Since N is original present in F-CN and F modification is proposed to induce the change of reactive sites from C atoms in CN to N atoms adjacent to the C-F bond in F-CN, a fundamental discussion on the mechanism should be given out. The change of energy barrier from theoretical simulation is more like phenomenon other than underlying comprehension.

Response: Thanks for your valuable comment. We have added a fundamental discussion on the mechanism in the DFT calculation part: Moreover, after the F atom occupies the C site, the C site is saturated, which becomes inert to the reactant or intermediate. As a result, the neighboring two-coordinated N site is the sole catalytic center on F-CN. The obtained reaction pathway shows that the surface N site in F-CN owns a much lower energy barrier (1.58 eV) than the surface C (2.25 eV) or N (2.86 eV) site in pristine CN (Fig. 3c), demonstrating that the F modification indeed can improve the OER activity on F-CN. According to the calculated charge density difference mappings, the improved OER activity on F-CN is attributed to the more local charge distribution between CN surface and *OH intermediate, which greatly promotes the formation of *OH intermediate (Fig. 3e) and also effectively avoids the excessively strong C-O interaction (Fig. 3d) or weak N-O interaction (Supplementary Fig. 20b) in pristine CN. As a result, the F modification greatly decreases the formation energy of rate-determining *OH. It should be noted that the F modification also significantly promote the formation of *OOH, which is also a high-barrier reaction step in OER (Fig. 3c). This implies that the excessively stable *O intermediate in the form of C=O bond on pristine CN is difficult to be further converted into *OOH. As a result, CN with an observable IR signal of C=O during reaction owns a lower activity than F-CN, which completely coincides with our experimental observations. Furthermore, the PDOS calculation provides an electronic-scale insight into the improved OER activity on F-CN. The obtained results show that F modification enables the N 2p states to move upward the Fermi level (Fig. 3f), which can be attributed to the transfer of partial electrons from N site to F site through C site. The more positive N 2p states promise the N site with higher oxidizing activity and more uncaptured orbits for bonding with *OH intermediate. Thus, the optimized bonding behavior between *OH and F-CN surface contributes to an improved OER activity. Hence, the N site in F-CN is the main OER center. Please see the **Line 20 of Page 9** in the revised manuscript.

Q4: Why the upward of Fermi level can enhance the stability of formed *OH on the N site in F-CN? Characterizations should be carried to support the that "Fermi level moved upward".

Response: Thanks for your valuable comment. Directly identifying the relationship between electronic states and catalytic activity is always a challenge in the catalytic field. However, DFT calculation is considered as an alternative tool to study it. We further explain the reason for the enhanced stability of *OH on the N site of F-CN in the revised manuscript: PDOS calculation provides an electronic-scale insight into the improved OER activity on F-CN. The obtained results show that F modification enables the N 2p states to move upward the Fermi level (Fig. 3f), which is attributed to the transfer of partial electrons from N site to F site through C site. The more positive N 2p states promise the N site with higher oxidizing activity and more uncaptured orbits for bonding with *OH intermediate. Thus, the optimized bonding behavior between *OH and F-CN surface contributes to an improved OER activity. Please see the **Line 15 of Page 10** in the revised manuscript.

Q5: Fig. 2c shows that the main activity difference of CN and F-CN lies in UV part. This conflicts with the pathway change proposed by the authors, because it is hardly to image that O₂ evolution proceeds in different ways under UV and visible light on F-CN.

Response: Thanks for your valuable comment. The activity of F-CN is higher than pristine CN catalyst under both UV and visible-light irradiation. **Fig. 2c** shows that the AQYs at 365 nm were determined to be 0.5718% on F_{0.1}-CN and 0.1281% on CN. AQYs at 420 nm on both samples were determined to be 0.0164% on F_{0.1}-CN and 0.0005% on CN. The relative activity difference between F_{0.1}-CN and CN is much larger in the visible-light region rather than in UV. However, since the CN sample was more active under UV, the absolute activity difference appeared to be larger. The OER mechanism should not be changed under UV and visible light.

Q6: It seems that H₂/O₂ production ratio on F-CN catalysts was larger than the stoichiometric ratio even taking account of the formation of H₂O₂. Why?

Response: Thanks for your valuable comment. Due to the inevitable generation of H₂O₂, the UV-induced conversion of O₂ to ozone, and the presence of dissolved O₂, it is reasonable that the H₂/O₂ ratio for the overall water splitting on semiconductor photocatalyst to be larger than a perfect stoichiometric ratio of 2:1 under white light. Similar phenomena were also observed in many other research works (Bai Y, et al, *Angew. Chem. Int. Ed.* 134, e202201299 (2022)). In our system, the main reason for the shortage of O₂ production is the formation of H₂O₂, since CN is well-acknowledged as an efficient catalyst for H₂O₂ production (Nat. Commun. 12, 3701 (2021); ACS Catalysis 10, 14380-14389 (2020)). The I⁻ ion titration method we previously used is not sensitive enough and could only qualitatively analyze the presence of H₂O₂. To quantify the H₂O₂ production during the reaction, we have employed a more sensitive Ce⁴⁺ back titration method (as reported in *Adv. Mater.* 34, e2107480 (2022)) (detailed methods see Supporting Information). As shown in **Fig. R1**, the H₂O₂ production rate on the champion F_{0.1}-CN catalyst was determined to be 85.36 $\mu\text{mol}\cdot\text{g}^{-1}\cdot\text{h}^{-1}$, almost identical to the short of H₂:O₂ ratio. The above discussion and additional experimental data were added to the revised manuscript. Please see the **Line 32 of Page 7** in the revised manuscript.

Fig. R1. (a) UV-vis absorption spectra of Ce⁴⁺ solution in different concentrations (0.15 mM-0.8 mM); (b) The concentration standard curve of Ce⁴⁺ solution. (c) The H₂O₂ production profiles on F_{0.1}-CN determined by the back titration of Ce⁴⁺.

Q7: As we all know, there are two types of C atoms in CN. The authors are suggested to clarify which kind of C is modified by the F atom.

Response: Thanks for your valuable comment. As shown in the structural configuration (**Fig. R2**) of CN, there are two types of N (i.e., two coordinated pyridine N_{2c} and three coordinated graphite N_{3c}) and one type of C (i.e., three coordinated C_{3c}) in CN. F modification was on C_{3c} sites (blue circle).

Fig. R2. Simulated structural configuration of CN supercell (2×3).

Q8: Can the author give more direct and reliable evidence to prove the formation of C-F bond?

Response: Thanks for your valuable comment. We monitored C 1s and F 1s states on F-CN before and after hydrothermal treatment by XPS. Before hydrothermal treatment, F⁻ ions were adsorbed on the CN surface, resulting in a distinct shift to the C 1s state (Fig. R3a), indicating the formation of C-F interaction. Moreover, after the hydrothermal treatment, the F 1s state slightly shifts towards higher binding energy, demonstrating the charge transfer from F atoms to C atoms, which solidly proves the strong C-F interaction. The additional experimental data was added to the Supplementary Information. Please see Supplementary Fig. 3.

Fig. R3. XPS (a) C 1s and (b) F 1s spectra of CN and F-CN samples before and after the hydrothermal treatment.

Q9: Crystal defects including N vacancy (VN) in CN have been well studied, and their effects in photocatalysis have also been demonstrated. So, the authors are suggested to evaluate the possible effect of defects in this system.

Response: Thanks for your valuable comment. We quantified the C/N ratio and F ion content by using elemental analysis and ion chromatography. As shown in Table. R1, with the increase of F content, the C/N ratio was almost unchanged. C/N ratio in all samples was $\sim 65\%$, demonstrating the existence of C defects rather than N defects. The unchanged C/N ratio excludes defect states as the major effect of enhanced performance on F-CN. Table. R1 was added to the Supplementary Information.

Table. R1. Elemental analysis of CN, CN-E, and F-CN samples.

Sample	C/N (at%)	F⁻ (mg/Kg)	F/C (at%)	F/C_{surf} (at%)
CN	64.98	0	0	0
CN-E	65	0	0	0
F _{0.01} -CN	65.08	1288.1939	0.2753	1.7895
F _{0.05} -CN	65.22	1617.9717	0.2854	1.8551
F _{0.1} -CN	65.24	3547.5642	0.6270	4.0755
F _{0.5} -CN	65.29	5718.8241	1.0216	6.6404
F _{0.7} -CN	65.47	13684.4692	2.4370	15.8405
F ₁ -CN	65.52	61008.9655	11.4104	74.1676

To Reviewer 3:

This is a very interesting study aiming at exploring the critical issue of oxygen evolution reaction on graphitic carbon nitride. The authors claimed that the formation of C=O bonding during continuous photocatalytic overall water splitting is the origin of carbon nitride failing to produce oxygen without the addition of small molecule organics. DRIFTA and NAP-XPS were used to support the claims, while fluorinated CN catalyst was designed to show that avoiding the formation of C=O can lead to the evolution of oxygen on carbon nitride without a sacrificing reagent. DFT calculations were also used to provide theoretical evidence. In general, the paper was well prepared, and can be considered for publication after some consideration of following issues.

Response: Thanks for your great efforts in reviewing our manuscript. We appreciate your valuable comments and suggestions.

Q1: The fluorinated CN catalyst, i.e., F-CN, is a critical sample for the study, however the characterization is not sufficient. Also, AFM, photocurrent, EIS should be performed for all the samples in study.

Response: Thanks for your valuable comment. As requested, we have performed more characterizations on F-CN samples (F_{0.1}-F₁), including elemental analysis (Table. R1), XRD (Fig. R1a), UV-vis DRS (Fig. R1b), BET surface area (Fig. R1c), AFM (Fig. R2), EIS (Fig. R3a), and photocurrent (Fig. R3b) measurements on F-CN catalysts with different F contents. All these characterizations and corresponding descriptions were added to the revised manuscript. Please see the Line 5 of Page 7 in the revised manuscript.

Table. R1. Elemental analysis of CN, CN-E, and F-CN samples.

Sample	C/N (at%)	F ⁻ (mg/Kg)	F/C (at%)	F/C _{surf} (at%)
CN	64.98	0	0	0
CN-E	65	0	0	0
F _{0.01} -CN	65.08	1288.1939	0.2753	1.7895
F _{0.05} -CN	65.22	1617.9717	0.2854	1.8551
F _{0.1} -CN	65.24	3547.5642	0.6270	4.0755
F _{0.5} -CN	65.29	5718.8241	1.0216	6.6404
F _{0.7} -CN	65.47	13684.4692	2.4370	15.8405
F ₁ -CN	65.52	61008.9655	11.4104	74.1676

Fig. R1. (a) XRD patterns, (b) UV-vis DRS spectra, and (c) N₂ adsorption-desorption isotherms of CN, CN-E and F-CN samples with different F contents. Inset of (c) shows the BET surface area of different samples.

Fig. R2. AFM images and corresponding line scan to determine the thickness of (a) CN, (b) CN-E, and (c) F_{0.1}-CN.

Fig. R3. (a) EIS and (b) photocurrent curves of CN-E, F_{0.1}-CN, and F₁-CN sample.

Q2: The chemical state of fluorine on the sample should be carefully explained. The XPS F1s spectra (Fig S3(b)) only show very weak signals with a high s/n ratio. The other concern is how the fluorine evolves after the reaction, given the fact that oxygen and hydrogen peroxide would appear.

Response: Thanks for your valuable comment. We have re-tested the XPS F1s spectra on the F_{0.1}-CN sample before and after the reaction. As shown in **Fig. R4**, the position of the F1s state was almost unchanged after the reaction. Please see the **Line 3 of Page 8** in the revised manuscript.

Fig. R4. The comparison of XPS (a) C1s and (b) F1s spectra on F_{0.1}-CN before and after 5 hrs of reaction.

Q3: The stability of gases production, and the properties of the photocatalysts should be checked.

Response: Thanks for your valuable comment. As shown in **Fig. R5**, we further extended the reaction time to 40 h, and the F_{0.1}-CN can still maintain 70.35% of the initial efficiency on H₂ and O₂ production and continue to work.

Fig. R5. Time course of the white-light photocatalytic overall water splitting on F_{0.1}-CN in 40 hrs.

Q4: If just aiming at the production of oxygen, other methods like loading cocatalyst on carbon nitride can also play the same role. What is the necessity of fluorination? How is this compared to other methods? How can this approach inspire or help other scientific challenges in materials chemical or science?

Response: Thanks for your valuable comment. Compared with metal-based inorganic semiconductor catalysts, metal-free catalysts have irreplaceable advantages in cost-control and environmental friendliness. CN is essentially a metal-free inorganic catalyst, however, due to the limitation of its interfacial reaction energy, it cannot work without metal/metal oxide cocatalysts. Therefore, meticulously studying the reaction mechanism at the CN interface to guide a completely metal-free CN-based photocatalyst is of great significance in terms of both cost and environmental friendliness.

To Reviewer 4:

This manuscript by Wu et al. reports a study on the reasons leading to the overall water splitting inactivation on single-phase CN catalysts. The authors claimed that the formation of inert C=O intermediates by oxidizing surface carbon atoms during the OER is the bottleneck for overall water splitting on g-C₃N₄ catalyst. After analyzing isotopic-labelled in-situ diffuse reflection infrared Fourier transform spectroscopy and in-situ near-ambient pressure X-ray photoelectron spectroscopy, the formation of C=O intermediates is prevented via a simple surface fluorination strategy, and the F_{0.1}-CN catalyst exhibits higher overall water splitting activity compared to the pristine CN. However, many mechanisms and modification strategies for g-C₃N₄ to improve the photocatalytic overall water splitting activity have been reported in previous literatures (Nat. Energy 6, 388(2021); Adv. Mater. 33, 2007479(2021); Chin. Chem. Lett. 32, 13(2021)). The novelty and significance of this work should be highlighted because the catalytic activity of F_{0.1}-CN catalyst is general compared to the catalysts reported so far. And some main conclusions are not well supported by the present experimental/theoretical results. I consider that the manuscript cannot meet the high standards of Nat. Commun., so I cannot recommend the publication of this work on the Nat. Commun.

Response: Thanks for your great efforts in reviewing our manuscript. We appreciate your valuable comments and suggestions.

Q1: Up to now, the nonmetallic-elements modification of g-C₃N₄-based catalysts and the corresponding catalytic mechanisms have been widely studied (Nat. Energy 6, 388(2021); Small 17, 2006851(2021); Adv. Mater. 33, 2007479(2021); Chin. Chem. Lett. 32, 13(2021)). Therefore, where are the advance and major breakthrough of the views mentioned by the author in relation to the reported results?

Response: Thanks for your valuable comment. In this work, rather than reporting a world-champion CN-based catalyst for overall water splitting, we aimed to figure out a fundamental scientific question of why the pristine CN surface is inert to OER, which is a long-puzzled scientific challenge for years since the discovery of CN catalysts.

Firstly, our *in-situ* DRIFTS and NAP-XPS provide valuable information at H₂O/CN interface to generate an in-depth understanding of this issue. After effectively changing the reaction path of OER on the CN surface by surface fluorination, we observed the occurrence of OER on the classical CN catalyst (without the help of OER cocatalyst), which is completely different from previously reported non-metallic modification strategies by changing the band structure, facilitating the interfacial charge transfer, or enhancing the light-harvesting ability. Especially, all these reported non-metallic modification strategies without exception still require the participation of metal oxide OER cocatalysts in the overall water splitting, which does not essentially solve the bottleneck problem that CN cannot induce OER on the surface.

Secondly, on the basis of these works, we are aware of the important role of the interfacial reaction pathway, using interfacial observation to confirm C=O as the reaction bottleneck, and then through surface fluorination, without changing the band structure and light absorption range, achieved the overall water splitting on F-CN catalysts.

Thirdly, our work reveals the reason why pristine CN cannot directly split water, and the overall water splitting can be achieved through a simple surface atomic modification strategy, which relieves the dependence of CN catalysts on metal-based OER cocatalysts and further enhances understanding of the interfacial reaction mechanism on CN at the atomic scale basis. Based on this, we think the novelty of our work meets the standard of Nature Communications. Thank you again for your valuable comment.

Q2: The F_{0.1}-CN catalyst exhibits poor overall water splitting activity (H₂ evolution rate of 174.77 μmol g⁻¹ h⁻¹) compared to the catalysts reported so far (H₂ evolution rate > 500 μmol g⁻¹ h⁻¹, λ > 420nm; Small 17, 2006851(2021); Adv.Mater. 32, 1907296(2020)). Using isotopic-labelled in-situ diffuse reflection infrared Fourier transform spectroscopy and in-situ near-ambient pressure X-ray photoelectron spectroscopy to study mechanism is of little significance.

Response: Thanks for your valuable comment. Our work realized the overall water splitting without metal-based OER cocatalysts, which reveals a long-hidden bottleneck of OER on the CN surface and provides a research basis for the future realization of completely metal-free CN catalysts. To reach the world-champion H₂ production efficiency is not the goal of

this work. Besides, the reported activity tests were not performed under a standard method. For example, the light intensity, irradiated area, mass of used catalyst can produce a significant effect on the activity. Hence, directly comparing the activities in different test methods cannot provide significant information on the performances of different photocatalysts. We have summarized the overall water splitting activities of recently reported CN-based catalysts, which are shown below in **Table R1**. Compared with reported CN catalysts, the overall water splitting efficiency on F-CN (our work) is not low. Moreover, ours is the only catalyst without severely changing the chemical composition or structure of CN, which is of great scientific significance.

Table. R1. Summary of g-C₃N₄-based materials for photocatalytic overall water splitting activity.

Catalysts	Co-catalyst	Mass (mg)	Light Source	H ₂ evolution rate (μmol g ⁻¹ h ⁻¹)	O ₂ evolution rate (μmol g ⁻¹ h ⁻¹)	Ref.
F-CN	3wt.% Pt	30	300 W Xe lamp, λ ≥ 300nm	174.77	44.15	This work
CNN/BDCNN	0.9wt.% Pt and 3wt.% Co(OH) ₂	40	300 W Xe lamp, λ ≥ 420nm	246.25	122	1
g-C ₃ N ₄ /rGO/PDIP	Pt/Cr ₂ O ₃ and Co(OH) ₂	50	300 W Xe lamp, λ ≥ 420nm	316	156	2
C ₃ N ₄ -rGO-WO ₃	1wt.% Pt	200	250 W metal halide lamp, λ ≥ 420nm	14.2	7.3	3
Co ₃ O ₄ /HCNS/Pt	1wt.% Pt	20	300 W Xe lamp, λ ≥ 300nm	155	75	4
α-Fe ₂ O ₃ /2D-C ₃ N ₄	3wt.% Pt and 0.1wt.% RuO ₂	50	300 W Xe lamp, λ ≥ 400nm	38.2	19.1	5
CoP/ g-C ₃ N ₄	3wt.% Pt	80	300 W Xe lamp, λ ≥ 300nm	250	125	6
TiO ₂ /g-C ₃ N ₄ -Ni(OH) ₂ / WO ₃	1wt.% Pt	200	150 W Xe lamp, λ ≥ 200nm	49	26	7
g-C ₃ N ₄ /BiVO ₄	3wt.% Pt	300	150 W Xe lamp, λ ≥ 395nm	36	18	8
MnO ₂ /g-C ₃ N ₄	3wt.% Pt	20	300 W Xe lamp, λ ≥ 400nm	60.5	29	9
CdSe/P-CN	1wt.% Pt	50	300 W Xe lamp, λ ≥ 420nm	113	55.6	10
PtMO _x /CN-M	Co ₃ O ₄	50	300 W Xe lamp, λ ≥ 420nm	47.6	22.8	11
CdS/Ni ₂ P/CN	-	50	300 W Xe lamp, λ ≥ 420nm	15.6	7.8	12
PCN/LaOCl	Pt and CoO _x	50	300 W Xe lamp, λ ≥ 420nm	160	76	13
CDots-C ₃ N ₄	-	80	300 W Xe lamp, λ ≥ 420nm	575	287.5	14
(C _{ring})-C ₃ N ₄	3wt.% Pt	30	300 W Xe lamp, λ ≥ 420nm	150	75	15
3D g-C ₃ N ₄ NS	1wt.% Pt and 3wt.% IrO ₂	50	300 W Xe lamp, λ ≥ 420nm	101.4	49.1	16
g-C ₃ N ₄ NWBs	1wt.% Pt	30	300 W Xe lamp, λ ≥ 300nm	72	35.6	17
TCN	Pt and RuO ₂	50	300 W Xe lamp, λ ≥ 350nm	110.4	44.8	18
CNSC	3wt.% Pt	25	300 W Xe lamp, λ ≥ 420nm	41.6	20.4	19

- Zhao D, et al. Boron-doped nitrogen-deficient carbon nitride-based Z-scheme heterostructures for photocatalytic overall water splitting. *Nature Energy* 6, 388-397 (2021).
- She X, et al. High Efficiency Photocatalytic Water Splitting Using 2D α-Fe₂O₃/g-C₃N₄ Z-Scheme Catalysts. *Advanced Energy Materials* 7, (2017).
- Chen X, Wang J, Chai Y, Zhang Z, Zhu Y. Efficient Photocatalytic Overall Water Splitting Induced by the Giant Internal Electric Field of a g-C₃N₄/rGO/PDIP Z-Scheme Heterojunction. *Adv Mater* 33, e2007479 (2021).
- Zhao G, Huang X, Fina F, Zhang G, Irvine JTS. Facile structure design based on C₃N₄ for mediator-free Z-scheme water splitting under visible light. *Catalysis Science & Technology* 5, 3416-3422 (2015).

- Zheng D, Cao XN, Wang X. Precise Formation of a Hollow Carbon Nitride Structure with a Janus Surface To Promote Water Splitting by Photoredox Catalysis. *Angew . Chem. Int . Ed.* 55, 11512-11516 (2016).
- Pan Z, Zheng Y, Guo F, Niu P, Wang X. Decorating CoP and Pt Nanoparticles on Graphitic Carbon Nitride Nanosheets to Promote Overall Water Splitting by Conjugated Polymers. *ChemSusChem* 10, 87-90 (2017).
- Yan J, Wu H, Chen H, Zhang Y, Zhang F, Liu SF. Fabrication of TiO₂/C₃N₄ heterostructure for enhanced photocatalytic Z-scheme overall water splitting. *Applied Catalysis B: Environmental* 191, 130-137 (2016).
- Martin DJ, Reardon PJ, Moniz SJ, Tang J. Visible light-driven pure water splitting by a nature-inspired organic semiconductor-based system. *J Am Chem Soc* 136, 12568-12571 (2014).
- Mo Z, et al. Self-assembled synthesis of defect-engineered graphitic carbon nitride nanotubes for efficient conversion of solar energy. *Applied Catalysis B: Environmental* 225, 154-161 (2018).
- Raziq F, et al. Photocatalytic solar fuel production and environmental remediation through experimental and DFT based research on CdSe-QDs-coupled P-doped-g-C₃N₄ composites. *Applied Catalysis B: Environmental* 270, (2020).
- Zeng Z, et al. Alkali-metal-oxides coated ultrasmall Pt sub-nanoparticles loading on intercalated carbon nitride: Enhanced charge interlayer transportation and suppressed backwark reaction for overall water splitting. *Journal of Catalysis* 377, 72-80 (2019).
- He H, et al. Distinctive ternary CdS/Ni₂P/g-C₃N₄ composite for overall water splitting: Ni₂P accelerating separation of photocarriers. *Applied Catalysis B: Environmental* 249, 246-256 (2019).
- Lin Y, Su W, Wang X, Fu X, Wang X. LaOCl-Coupled Polymeric Carbon Nitride for Overall Water Splitting through a One-Photon Excitation Pathway. *Angew Chem Int Ed Engl* 59, 20919-20923 (2020).
- Liu J, et al. Water splitting. Metal-free efficient photocatalyst for stable visible water splitting via a two-electron pathway. *Science* 347, 970-974 (2015).
- Che W, et al. Fast Photoelectron Transfer in (C_{ring})-C₃N₄ Plane Heterostructural Nanosheets for Overall Water Splitting. *J Am Chem Soc* 139, 3021-3026 (2017).
- Chen X, et al. Three-dimensional porous g-C₃N₄ for highly efficient photocatalytic overall water splitting. *Nano Energy* 59, 644-650 (2019).
- Zhang K, et al. Tunable Bandgap Energy and Promotion of H₂O₂ Oxidation for Overall Water Splitting from Carbon Nitride Nanowire Bundles. *Advanced Energy Materials* 6, (2016).
- Chen L, et al. Graphitic Carbon Nitride Microtubes for Efficient Photocatalytic Overall Water Splitting: The Morphology Derived Electrical Field Enhancement. *ACS Sustainable Chemistry & Engineering* 8, 14386-14396 (2020).
- Zeng Y, et al. Sea-urchin-structure g-C₃N₄ with narrow bandgap (~2.0 eV) for efficient overall water splitting under visible light irradiation. *Applied Catalysis B: Environmental* 249, 275-281 (2019).

Q3: According to the signal-to-noise ratio of F1s spectra, the content of F should be very low for the F_{0.1}-CN catalyst (< 0.1%). The authors believed that the improved OER activity is due to the C-F bond formation optimizing the OER pathway on adjacent N atoms. However, the number of F-modified N sites should be minimal. The coordination environment of most carbon sites in F_{0.1}-CN is similar to that in pristine CN. Therefore, it is not convincing that F-CN does not generate C=O intermediates during the reaction. The authors should provide more experimental and theoretical results to explain this issue in detail.

Response: Thanks for your valuable comment. We quantified the C/N ratio and F ion content by using elemental analysis and ion chromatography. The F content and calculated surface F coverage (the number of exposed C atoms on CN surface were calculated by the thickness of the sample, measured by AFM) were summarized in **Table. R2**. For the champion F_{0.1}-CN sample, the surface F coverage is about 4 atom%. DFT calculations with varying F contents (with surface F coverage in the range of 1~8 atom%) were also conducted. As shown in **Fig. R1**, with surface F coverage increases from 1 atom% to 8 atom%, the barrier of OER on the adjacent N site (**Fig. R1a**) and the energy level of N 2p state (**Fig. R1b**) are almost unchanged, indicating that the adjacent N site of the C-F structure is only affected by the adjacent C-F hybridization rather than the concentration of F. The F coverage only affects the number of active N sites. Based on the above experimental and calculation results, we reason that once the active N site emerged, water splitting reaction has a higher chance of occurring at the active N site adjacent to C-F rather than at normal C or N sites, by which hinders the surface C=O formation. The additional experimental data and discussions were added to the revised manuscript. Please see the **Line 6 of Page 10** and **Line 21 of Page 10** in the revised manuscript.

Table. R2. Elemental analysis of CN, CN-E, and F-CN samples.

Sample	C/N (at%)	F ⁻ (mg/Kg)	F/C (at%)	F/C _{surf} (at%)
CN	64.98	0	0	0
CN-E	65	0	0	0
F _{0.01} -CN	65.08	1288.1939	0.2753	1.7895
F _{0.05} -CN	65.22	1617.9717	0.2854	1.8551
F _{0.1} -CN	65.24	3547.5642	0.6270	4.0755
F _{0.5} -CN	65.29	5718.8241	1.0216	6.6404
F _{0.7} -CN	65.47	13684.4692	2.4370	15.8405
F ₁ -CN	65.52	61008.9655	11.4104	74.1676

Fig. R1. (a) Free energy profiles of OER on F-CN (N) with F surface coverage of 1 atom% to 8 atom% (denoted F1~F8). F-CN (N) represents N reaction sites on F-CN (C reaction sites occupied entirely by F atoms). (b) PDOS of N 2p states in F-CN with F surface coverage of 1 atom% to 8 atom% (denoted F1~F8). The dashed line stands for Fermi level.

Q4: In the in-situ NAP-XPS observations of N1s spectra, the peak area ratios of NH_x, NC₃, C-N=C change significantly for the pristine CN and F_{0.1}-CN catalysts. This phenomenon should be explained in detail. In addition, the authors claimed that the N site in F-CN is the main OER active center. However, during continuous white light illumination, there is no peak positions change or new peaks appear on the N1s spectra for the F_{0.1}-CN catalyst, which is unreasonable.

Response: Thanks for your valuable comment. Firstly, in CN, the electron orbitals of C and N atoms are highly hybridized. After the F modification, the electron orbital of F is hybridized with C atoms, by which affects the electron orbital of N atoms. However, during the reaction, the unchanged N 1s state in F-CN indicates that the charge transfer at the N site was very fast with no obvious accumulation of intermediate states. This result further suggests that the N site as the main OER center facilitates the interfacial charge transfer after the surface F-modification, consistent with our hypothesis.

Q5: In the in-situ DRIFTS analysis, the authors claimed that the O source of C=O is from H₂O. However, an obvious peak of the C=¹⁶O stretching vibration is still observed at 1725 cm⁻¹ when H₂¹⁶O is replaced by ¹⁸O-labelled H₂¹⁸O. Furthermore, why does the bending vibration of H-O-H at 1645 cm⁻¹ disappear in Fig. 1b-1d? More detailed analysis should be provided.

Response: Thanks for your valuable comment. First, we think the presence of C=O¹⁶ peak in H₂O¹⁸/CN system is due to the inevitable atomic exchange between the abundant surface -OH groups on CN surface and H₂O¹⁸. Furthermore, the reason why the bending vibration of H-O-H at 1645 cm⁻¹ was unobservable is that its position coincides with the positive C=O¹⁸ peak.

Q6: In the DFT calculations, the authors only compared the energy barrier of the surface N sites of F-CN and the surface C and N sites of pristine CN. Why did the authors not consider the C atom of the C-F bond in F-CN as the reaction site?

Response: Thanks for your valuable comment. As shown in the structural configuration (Fig. R2) of CN, there are two types of N (i.e., two coordinated N_{2c} and three coordinated N_{3c}) and one type of C (i.e., three coordinated C_{3c}) in CN. F modification was on C_{3c} sites (blue circle) to reach the maximum coordination (C_{4c}), which is unable to bind with other reactant molecules.

Fig. R2. Simulated structural configuration of CN supercell (2×3).

Q7: The authors should improve the accuracy of the expression. For example, what is the wavelength of white light?

Response: Thanks for your valuable comment. We have improved the accuracy of expressions throughout the manuscript as requested. The white light source is a 300W Xenon light (>300 nm) with the spot center intensity of 1000 mW/cm^2 .

To Reviewer 5:

In this manuscript, the authors observed a collective C=O bonding during continuous photocatalytic overall water splitting on g-C₃N₄ catalyst by in-situ DRIFTS and NAP-XPS, and confirmed that the inert C=O bond directly hinders further OER steps, resulting in negligible O generation on CN. The F_{0.1}-CN catalyst prepared based on this finding exhibited excellent overall water splitting activity with the order-of-magnitude improved H₂ evolution rate compared to the pristine CN catalyst. This manuscript was interesting. However, I feel that there are still a number of uncertainties and obvious problems. The analysis of the data is not accurate enough. After careful evaluation, I think that this work does not meet the standard of Nature Communications.

Response: Thanks for your great efforts in reviewing our manuscript. We appreciate your valuable comments and suggestions.

Q1: In-situ NAP-XPS is the main method for the authors to prove the formation and disappearance of C=O bonds. This method is very interesting, but I think there are some issues with data analysis on NAP-XPS. For example, in the XPS spectra of CN under different illumination times, the fitted peak width of the O-H peak changed significantly. Actually, the total spectra at 0 min and 5 min of CN are close, but the C=O peak is formed due to the change in the peak width of the O-H peak during fitting. Authors should fit the XPS data under the same O-H peak width to determine the presence of C=O peak.

Response: Thanks for your valuable comment. We have reanalyzed the NAP-XPS data by fitting all spectra under the same peak width. As shown in **Fig. R1**, after the fitting, the C=O formation can be clearly observed. During 0-5 min, the change in O1s spectra (532.7 eV) was also observable. We have added the reanalyzed result to the revised manuscript. Please see **Fig. 1e**.

Fig. R1. *In-situ* NAP-XPS O1s spectra on pristine CN with 0.2 mbar H₂O vapor pressure using a 300 W Xenon lamp as the white light source in 15 min.

Q2: The consumption of e⁻ and h⁺ during photocatalytic overall water splitting should be equal. However, in this work, the e⁻ consumed by H₂ production is not equal to the h⁺ consumed by O₂ and H₂O₂ production. It is suggested that the authors calculate the amount of reduction and oxidation products in the photocatalytic water splitting process and analyze why the consumption ratio of e⁻ and h⁺ is not 1:1.

Response: Thanks for your valuable comment. Due to the inevitable generation of H₂O₂, the UV-induced conversion of O₂ to ozone, and the presence of dissolved O₂, it is reasonable that the H₂/O₂ ratio for the overall water splitting on semiconductor photocatalyst to be larger than a perfect stoichiometric ratio of 2:1 under white light. Similar phenomena were also observed

in many other research works (Bai Y, et al, *Angew. Chem. Int. Ed.* 134, e202201299 (2022)). In our system, the main reason for the shortage of O₂ production is the formation of H₂O₂, since CN is well-acknowledged as an efficient catalyst for H₂O₂ production (*Nat. Commun.* 12, 3701 (2021); *ACS Catalysis* 10, 14380-14389 (2020)). The I⁻ ion titration method we previously used is not sensitive enough and could only qualitatively analyze the presence of H₂O₂. To quantify the H₂O₂ production during the reaction, we have employed a more sensitive Ce⁴⁺ back titration method (as reported in *Adv. Mater.* 34, e2107480 (2022).) (detailed methods see Supporting Information). As shown in **Fig. R2**, the H₂O₂ production rate on the champion F_{0.1}-CN catalyst was determined to be 85.36 μmol·g⁻¹·h⁻¹, almost identical to the short of H₂:O₂ ratio. The above discussion and additional experimental data were added to the revised manuscript. Please see the **Line 32 of Page 7** in the revised manuscript.

Fig. R2. (a) UV-vis absorption spectra of Ce⁴⁺ solution in different concentrations (0.15 mM-0.8 mM); (b) The concentration standard curve of Ce⁴⁺ solution. (c) The H₂O₂ production profiles on F_{0.1}-CN determined by the back titration of Ce⁴⁺.

Q3: The authors should point out the test conditions of Supplementary Figure 9, white light or AM1.5G?

Response: Thanks for your valuable comment. The H₂O₂ production was measured under white light irradiation (Supplementary Figure 9). The test conditions were described in the Supplementary Method section.

Q4: The authors aim to illustrate the effect of F doping on enhanced OER performance, then supplemental separate OER tests may be more helpful in illustrating the authors' hypothesis.

Response: Thanks for your valuable comment. We have performed the electrochemical OER experiment on CN, CN-E, and F_{0.1}-CN samples. The result shows that the F-modification indeed reduces the OER potential of F-CN in comparison with CN and CN-E (**Fig. R3**), consistent with our hypothesis. Please see the **Line 34 of Page 8** in the revised manuscript.

Fig. R3. Polarization curves of CN, CN-E and F_{0.1}-CN samples.

Q5: The nanosheets of CN in Supplementary Figure 1 appear thinner than F-CN, but its BET surface is much smaller than that of F-CN. What is the reason for the increase in the specific surface area of F-CN?

Response: Thanks for your valuable comment. TEM images (Supplementary Figure 1) can only distinguish local morphologies of the dispersed samples, which is difficult to intuitively identify the specific thickness of the sample. We thusly

employed AFM to characterize the thickness of CN, CN-E, and F-CN samples. As shown in Fig. R4, CN-E and F-CN samples have a smaller layer thickness of ~ 4 nm, whereas the pristine CN layer is about ~ 5 nm. The increased BET surface area of F-CN compared to CN is due to the reduced layer thickness from the hydrothermal exfoliation. The additional experimental data was added to the Supplementary Information. Please see Supplementary Fig. 7.

Fig. R4. AFM images and corresponding line scan to determine the thickness of (a) CN, (b) CN-E, and (c) F_{0.1}-CN.

Q6: It is difficult to illustrate the existence of F in Supplementary Figure 3. It is recommended that the authors re-test XPS and consider changing the sample preparation method to improve the accuracy.

Response: Thanks for your valuable comment. We have retested XPS F1s spectra on CN, and F-CN samples before and after hydrothermal treatment. The existence of F is now observable (Fig. R5). The additional experimental data was added to the Supplementary Information. Please see Supplementary Fig. 3.

Fig. R5. XPS (a) C1s and (b) F1s spectra of CN and F-CN samples before and after the hydrothermal treatment.

Reviewers' Comments:

Reviewer #1:

Remarks to the Author:

According to the responses, the authors made basic revisions based on the comments, but did not fundamentally solve the reviewer's doubts. Especially the reliability of in situ DRIFTS and NAP-XPS signals for identifying the surface C=O bonds. In my opinion, the revision of the paper does not reach the level of publication in this journal; unfortunately, I suggest the rejection of this manuscript.

1. Although the authors prove that the hydrophilicity of F-CN is greatly decreased with the increase of F content, this could not be an objective reason for the decreased OER activity. Otherwise, the authors need to cite references to demonstrate the relationship between hydrophilicity and OER performance.

2. The authors need to provide more direct evidence such as absorption spectra, to prove and reveal the coordination environment of C-F, not just through simulation calculation.

3. The authors still do not explain the reliability of in situ DRIFTS to identify C=O intermediate formation. The in-situ test here cannot define as the real in situ conditions during the reaction, considering the pressure and temperature (150°C) discrepancy from the actual water splitting test. Also, the authors used a 420nm LED light source, which is totally different from the light source during the water splitting test (≥ 300 nm, 1000 mW·cm⁻²). Did the author add H₂PtCl₆·6H₂O during the in situ DRIFTS test? If not, why? All of these will affect the analysis of the surface reaction mechanism.

4. The authors do not have a good explanation of what the merits of F-CN is. Especially compared with some benchmark non-metallic element doped or complex modified catalysts (Nature Energy 6, 388-397 (2021); Advanced Energy Materials 7, (2017), Science 347, 970- 974 (2015)). In addition, the authors used the UV-vis light irradiation with the high light intensity (1000 mW/cm²) and the high concentrated Pt cocatalyst for the reaction, which has no advantage. Therefore, from this point of view, the scientific importance of studying this material is not very significant.

5. The authors did not consider the effect of Pt cocatalyst on the F-CN. It was claimed that the adjacent N site of the C-F structure is responsible for the OER test; what is the reason for the largely enhanced HER half-reaction.

6. The stoichiometric ratio of hydrogen to oxygen production is far from two to one, which is crucial for studying fully decomposed water. The authors need to better illustrate this point by calculating Faraday efficiency or STH. In addition, the author can make a graph or bar chart with the hydrogen/oxygen ratio as the ordinate and the fluorine doping amount as the abscissa to study the influence of fluorine doping.

7. As far as I know, this journal has high requirements for repeatability of data, especially for statistics figures, such as Fig.2 a (the most important performance data I think); the error bars are usually needed and defined thoroughly.

8. The captions in supplementary Figure 5 are inconsistent with the figure curves.

Reviewer #2:

Remarks to the Author:

Though efforts have been taken by the authors, part of the comments have not been well responded:

1、 Q1 ("In Fig. 3c, the rate-determining step of O₂ evolution over CN(C) is the formation of *OH rather than the step from O* to *OOH. So, why the formation of C=O is the bottleneck of overall water splitting on CN-based catalysts? More illustration or an additional attention on the effect of C=O

should be given out.") is not properly responded.

It seems that they authors do not make clear in the manuscript that what is "the formation of C=O". Is it the same with "the step from *OH to O*"? Even taking consideration of this point, some important even crucial conclusions are inconsistent with the DFT results. It can be seen from Fig. 3c that, the formation energy barrier of *OH is 2.25 eV and the energy barrier of O* to *OOH is about 2 eV while the corresponding value of the formation of O* or C=O is about 0.85, which indicates the rate-determining step of O₂ evolution over CN(C) is the formation of *OH. This conflicts with the authors conclusion that "formation of C=O intermediate is an important bottleneck for overall water splitting on single-phased CN" and even the title "Intermediate C=O Formation Is the Bottleneck of Overall Water Splitting on Carbon Nitride".

The authors are recommend to carefully analysis the results and reorganize the manuscript.

2、 The answer to Q2 (The authors proposed that the N site is the main OER center in F-CN. While N 1s XPS spectra indicates no obvious shift under continuous white light illumination. Why an oxidized N state did not form like C in CN?) is unconvincing. From the results of DFT, the energy barrier of CN(C) and F-CN(N) has a similar trend. So, the statement that "charge transfer at the N site was very fast with no obvious accumulation of intermediate states" lead to the unchanged N 1s state in F-CN is confusing. Please offer a more reasonable explanation.

3、 The theoretical O₂ production of CN is very low and may exceed the minimum detection value of TCD. The original GC spectra of pristine CN photocatalytic water splitting (e.g., 5h results of Fig. 2b) should be given.

4、 For Q4, we advise the authors to clarify the stability of *OH and the adsorption strength of *OH. Is it possible to use electrochemical methanol oxidation experiment to offer further evidence?

5、 For Q7, please carefully check the reference of CN materials (e.g., Angew. Chem. Int. Ed. 2018, 57, 6848–6852) and get a basic understanding about its structure. Are you sure "there are two types of N one type of C"?

Reviewer #3:

Remarks to the Author:

The revision is satisfactory.

Reviewer #4:

Remarks to the Author:

In the revised manuscript NCOMMS-22-17621A, the authors highlighted the innovation of this work compared to previous reported similar works, and conducted a systematic comparison of overall water splitting activities with recently reported CN-based catalysts. The authors have added some experimental results (AFM, EIS and Quantification of H₂O₂) to support the structure and reaction mechanism of F-CN according to the reviewers' suggestions. The quality of the manuscript has been improved significantly, and now I recommend the publication of this work in Nature Communication. However, the related experimental results are still controversial. The authors should address the following issues.

1. The authors claimed that no changes are found on the N1s spectra on both CN and F0.1-CN samples under the white light illumination. However, the peak area ratios of NH_x, NC₃, C-N=C change significantly. The authors believed that the hybridization of F and C atoms affects the electron orbital of N atoms. This is not sufficient to explain the changes in the in situ NAP-XPS. In addition, detailed results of the XPS fit parameters should be provided.

2. The F0.1-CN catalyst exhibited excellent overall water splitting capacity with H₂ evolution rate of 83.89 μmol·g⁻¹·h⁻¹ and O₂ evolution rate of 21.15 μmol·g⁻¹·h⁻¹. The authors employed a Ce⁴⁺ back titration method to quantify the H₂O₂ production during the reaction. The H₂O₂ production rate on the champion F0.1-CN catalyst was determined to be 85.36 μmol·g⁻¹·h⁻¹. Considering the formation of H₂O₂, it seems that the H₂/O₂ production ratio on the F0.1-CN catalyst would be less than the

stoichiometric ratio of 2:1. Why?

3. In Supplementary Fig. 9b, the authors should clearly note that the UV-vis absorption of the F-CN catalyst was slightly blue-shifted with high F content.

Reviewer #5:

Remarks to the Author:

In this manuscript, the authors used in-situ FTIR and NAP-XPS with isotopic detection to reveal that C=O formation is the inhibiting factor on carbon nitride for photocatalytic overall water splitting to form H₂ and O₂. A surface fluorination strategy was employed to improve the OER performance by occupying carbon-sites on CN with F-ions. This work is insightful and can be published after revision.

1) Why the authors choose fluorination strategy to inhibit the formation of C=O? There could be many other choices (such like sulfuration or phosphorization) to occupy the C sites of CN, but the authors seemed to go to fluorination without showing the underlying reason.

2) To illustrate the effect of F on enhancing the OER performance, the authors should show more detailed OER test rather than just the LSV. To understand why CN is inert to OER and why F-CN is conducive to OER, the OER kinetics analysis would be much more important and inspiring than simply knowing the onset potential.

Reply to reviewers' comments

To Reviewer 1:

According to the responses, the authors made basic revisions based on the comments, but did not fundamentally solve the reviewer's doubts. Especially the reliability of in situ DRIFTS and NAP-XPS signals for identifying the surface C=O bonds. In my opinion, the revision of the paper does not reach the level of publication in this journal; unfortunately, I suggest the rejection of this manuscript.

Response: Thanks for your great efforts in reviewing our manuscript. We appreciate your valuable comments and suggestions.

Q1: Although the authors prove that the hydrophilicity of F-CN is greatly decreased with the increase of F content, this could not be an objective reason for the decreased OER activity. Otherwise, the authors need to cite references to demonstrate the relationship between hydrophilicity and OER performance.

Response: Thanks for your valuable comment. The decreased photocatalytic activity of F-CN with high F content is not directly affected by the increased hydrophilicity. As we demonstrated in Fig. R1, hydrophilicity affects the sedimentation behavior of catalysts in water. With high F content, the F-CN catalyst can sediment very fast and be very difficult to form a uniform aqueous suspension. Such a fast sedimentation behavior of the catalyst will definitely render a lower light absorption efficiency than the well-suspended counterpart. In addition, we employed the *in-situ* UV-vis optical fiber spectroscopy to directly monitor the transmittance of white light (tungsten lamp, 5W) through different suspensions (0.3g/L). As shown in Fig. R2, with the increasing F content in F-CN catalysts ($F_{0.01}$ ~ F_1), the transmittance of white light is significantly increased (30.03%~63.79%), demonstrating that more light is transmitted with higher F content, therefore reducing the photocatalytic performance. For the corresponding description in the revised manuscript see Page 7 Line 24.

Fig. R1. (a) The comparison of continuous sedimentation of CN-E and F-CN samples with different F content, and water surface contact angle on (b) CN-E, (c) F_{0.01}-CN, (d) F_{0.05}-CN, (e) F_{0.1}-CN, and (f) F₁-CN.

Fig. R2. Light transmittance through different catalyst suspensions (0.3g/L) by in situ optical fiber spectroscopy (under constant 500 rpm magnetic stirring). The white light source is a tungsten lamp (5W). Water is the transparent solvent.

Q2: The authors need to provide more direct evidence such as absorption spectra, to prove and reveal the coordination environment of C-F, not just through simulation calculation.

Response: Thanks for your valuable comment. We performed the IR measurement on CN and F-CN samples as shown in **Fig. R3**. Unfortunately, corresponding C-F vibrations were not observed due to the lower content of surface C-F in F-CN samples. We further employed XPS C1s and F1s spectra (**Fig. R4**) to monitor the strong C-F interaction in F-CN samples. Before the hydrothermal treatment, with the emerging of the F 1s state, a distinct shift of the C 1s state was observed (**Fig. R4a**), demonstrating the interaction between C and F atoms. Moreover, after the hydrothermal treatment, the F 1s state slightly shifts towards higher binding energy (**Fig. R4b**), demonstrating the charge transfer from F atoms after hydrothermal treatment, which solidly proves the strong C-F interaction in F-CN.

Fig. R3. Infrared absorption spectra of different catalysts.

Fig. R4. XPS (a) C1s and (b) F1s spectra of CN and F-CN samples before and after the hydrothermal treatment.

Q3: The authors still do not explain the reliability of in situ DRIFTS to identify C=O intermediate formation. The in-situ test here cannot define as the real in situ conditions during the reaction, considering the pressure and temperature (150°C) discrepancy from the actual water splitting test. Also, the authors used a 420nm LED light source, which is totally different from the light source during the water splitting test (≥ 300 nm, 1000 mW·cm⁻²). Did the author add H₂PtCl₆·6H₂O during the in situ DRIFTS test? If not, why? All of these will affect the analysis of the surface reaction mechanism.

Response: Thanks for your valuable comment. First, all *in-situ* DRIFTS experiments were at 25°C under ambient pressure conditions rather than at 150 °C. As described in the experimental method that “Prior to isotopic experiments, pristine CN and F-CN samples were heated at 423 K for 30 min under flowing N₂ to remove the remaining water.”, the 423 K treatment was only employed before the isotopically labeled measurement of CN/F-CN samples to eliminate residue water. After that, samples were cooled down to room temperature and measured under identical conditions as other unlabeled

experiments. A more detailed description of the experimental methods can be found in the revised Supplementary Information (See SI, Page 2, Line 10).

Second, CN is a visible-light-driven catalyst. Therefore, we used the 420 nm LED lamp as the light source in the *in-situ* DRIFTS experiments. As requested, we further conducted the *in-situ* DRIFTS experiments under white light irradiation (Xe lamp, ≥ 300 nm, center intensity of $1000 \text{ mW}\cdot\text{cm}^{-2}$). As shown in **Fig. R5**, the experimental phenomenon is consistent with the use of a 420 nm lamp, indicating that the excitation light source does not affect the mechanism of the reaction on CN/F-CN.

Third, Pt was loaded on CN/F-CN to facilitate HER ($2\text{H}^+ + 2\text{e}^- \rightarrow \text{H}_2$) during the water splitting reaction, which has no IR absorption signals and should not affect the OER mechanism. Therefore, we did not consider Pt in DRIFTS experiments. As requested, we further conducted the *in-situ* DRIFTS experiments on CN/F-CN samples with the loading of Pt. As shown in **Fig. R6**, the experimental phenomenon is consistent with that on the pristine CN/F-CN samples, indicating that the Pt loading indeed does not affect the OER mechanism on CN/F-CN.

All these control experiments demonstrate the reliability of our *in-situ* DRIFTS results. Additional control experiments were added to the revised manuscript; Please see **Page 5, Line 13**.

Fig. R5. DRIFTS spectra *in-situ* monitored at (a) CN/H₂O and (b) F_{0.1}-CN/ H₂O interface under constant white light (Xe lamp, ≥ 300 nm) irradiation in 15 min.

Fig. R6. DRIFTS spectra *in-situ* monitored at (a) Pt/CN/H₂O and (b) Pt/F_{0.1}-CN/ H₂O interface under constant white light (Xe lamp, ≥ 300 nm) irradiation in 15 min.

Q4: The authors do not have a good explanation of what the merits of F-CN is. Especially compared with some benchmark non-metallic element doped or complex modified catalysts (Nature Energy 6, 388-397 (2021); Advanced Energy Materials 7,

(2017), Science 347, 970- 974 (2015)). In addition, the authors used the UV-vis light irradiation with the high light intensity (1000 mW/cm²) and the high concentrated Pt cocatalyst for the reaction, which has no advantage. Therefore, from this point of view, the scientific importance of studying this material is not very significant.

Response: Thanks for your valuable comment. In this work, rather than reporting a world-champion CN-based catalyst for overall water splitting, we aimed to figure out a fundamental scientific question of why the pristine CN surface is inert to OER, which is a long-puzzled scientific challenge for years since the discovery of CN catalysts. Our work realized the overall water splitting without metal-based OER cocatalysts, which provides a research basis for the future realization of completely metal-free CN catalysts. Reaching the world-champion H₂ production efficiency is not the goal of this work. Besides, the reported activity tests were not performed under a standard method. For example, the light intensity, irradiated area, and mass of the used catalyst can produce a significant effect on the activity. Hence, directly comparing the activities in different test methods cannot provide significant information on the performances of different photocatalysts. We have summarized the overall water-splitting activities of recently reported CN-based catalysts, which are shown below in **Table R1**. Compared with reported CN catalysts, the overall water splitting efficiency on F-CN (our work) is not low. Pt is the HER co-catalyst, which is almost necessary to all CN catalysts for water splitting. Moreover, ours is the only catalyst without severely changing the chemical composition or structure of CN, demonstrating the key role of surface reaction configurations, which is of great scientific significance.

Table. R1. Summary of g-C₃N₄-based materials for photocatalytic overall water splitting activity.

Catalysts	Co-catalyst	Mass (mg)	Light Source	H ₂ evolution rate (μmol g ⁻¹ h ⁻¹)	O ₂ evolution rate (μmol g ⁻¹ h ⁻¹)	Ref.
F-CN	3wt.% Pt	30	300 W Xe lamp, λ ≥ 300nm	174.77	44.15	This work
CNN/BDCNN	0.9wt.% Pt and 3wt.% Co(OH) ₂	40	300 W Xe lamp, λ ≥ 420nm	246.25	122	1
g-C ₃ N ₄ /rGO/PDIP	Pt/Cr ₂ O ₃ and Co(OH) ₂	50	300 W Xe lamp, λ ≥ 420nm	316	156	2
C ₃ N ₄ -rGO-WO ₃	1wt.% Pt	200	250 W metal halide lamp, λ ≥ 420nm	14.2	7.3	3
Co ₃ O ₄ /HCNS/Pt	1wt.% Pt	20	300 W Xe lamp, λ ≥ 300nm	155	75	4
α-Fe ₂ O ₃ /2D-C ₃ N ₄	3wt.% Pt and 0.1wt.% RuO ₂	50	300 W Xe lamp, λ ≥ 400nm	38.2	19.1	5
CoP/ g-C ₃ N ₄	3wt.% Pt	80	300 W Xe lamp, λ ≥ 300nm	250	125	6
TiO ₂ /g-C ₃ N ₄ -Ni(OH) ₂ / WO ₃	1wt.% Pt	200	150 W Xe lamp, λ ≥ 200nm	49	26	7
g-C ₃ N ₄ /BiVO ₄	3wt.% Pt	300	150 W Xe lamp, λ ≥ 395nm	36	18	8
MnO ₂ /g-C ₃ N ₄	3wt.% Pt	20	300 W Xe lamp, λ ≥ 400nm	60.5	29	9
CdSe/P-CN	1wt.% Pt	50	300 W Xe lamp, λ ≥ 420nm	113	55.6	10
PtMO _x /CN-M	Co ₃ O ₄	50	300 W Xe lamp, λ ≥ 420nm	47.6	22.8	11
CdS/Ni ₂ P/CN	-	50	300 W Xe lamp, λ ≥ 420nm	15.6	7.8	12
PCN/LaOCl	Pt and CoO _x	50	300 W Xe lamp, λ ≥ 420nm	160	76	13
CDots-C ₃ N ₄	-	80	300 W Xe lamp, λ ≥ 420nm	575	287.5	14
(C _{ring})-C ₃ N ₄	3wt.% Pt	30	300 W Xe lamp, λ ≥ 420nm	150	75	15
3D g-C ₃ N ₄ NS	1wt.% Pt and 3wt.% IrO ₂	50	300 W Xe lamp, λ ≥ 420nm	101.4	49.1	16
g-C ₃ N ₄ NWBs	1wt.% Pt	30	300 W Xe lamp, λ ≥ 300nm	72	35.6	17
TCN	Pt and RuO ₂	50	300 W Xe lamp,	110.4	44.8	18

			$\lambda \geq 350\text{nm}$			
CNSC	3wt.% Pt	25	300 W Xe lamp, $\lambda \geq 420\text{nm}$	41.6	20.4	19

- Zhao D, et al. Boron-doped nitrogen-deficient carbon nitride-based Z-scheme heterostructures for photocatalytic overall water splitting. *Nature Energy* 6, 388-397 (2021).
- She X, et al. High Efficiency Photocatalytic Water Splitting Using 2D $\alpha\text{-Fe}_2\text{O}_3/\text{g-C}_3\text{N}_4$ Z-Scheme Catalysts. *Advanced Energy Materials* 7, (2017).
- Chen X, Wang J, Chai Y, Zhang Z, Zhu Y. Efficient Photocatalytic Overall Water Splitting Induced by the Giant Internal Electric Field of a $\text{g-C}_3\text{N}_4/\text{rGO}/\text{PDIP}$ Z-Scheme Heterojunction. *Adv Mater* 33, e2007479 (2021).
- Zhao G, Huang X, Fina F, Zhang G, Irvine JTS. Facile structure design based on C_3N_4 for mediator-free Z-scheme water splitting under visible light. *Catalysis Science & Technology* 5, 3416-3422 (2015).
- Zheng D, Cao XN, Wang X. Precise Formation of a Hollow Carbon Nitride Structure with a Janus Surface To Promote Water Splitting by Photoredox Catalysis. *Angew . Chem. Int . Ed.* 55, 11512-11516 (2016).
- Pan Z, Zheng Y, Guo F, Niu P, Wang X. Decorating CoP and Pt Nanoparticles on Graphitic Carbon Nitride Nanosheets to Promote Overall Water Splitting by Conjugated Polymers. *ChemSusChem* 10, 87-90 (2017).
- Yan J, Wu H, Chen H, Zhang Y, Zhang F, Liu SF. Fabrication of $\text{TiO}_2/\text{C}_3\text{N}_4$ heterostructure for enhanced photocatalytic Z-scheme overall water splitting. *Applied Catalysis B: Environmental* 191, 130-137 (2016).
- Martin DJ, Reardon PJ, Moniz SJ, Tang J. Visible light-driven pure water splitting by a nature-inspired organic semiconductor-based system. *J Am Chem Soc* 136, 12568-12571 (2014).
- Mo Z, et al. Self-assembled synthesis of defect-engineered graphitic carbon nitride nanotubes for efficient conversion of solar energy. *Applied Catalysis B: Environmental* 225, 154-161 (2018).
- Raziq F, et al. Photocatalytic solar fuel production and environmental remediation through experimental and DFT based research on CdSe-QDs-coupled P-doped-g- C_3N_4 composites. *Applied Catalysis B: Environmental* 270, (2020).
- Zeng Z, et al. Alkali-metal-oxides coated ultrasmall Pt sub-nanoparticles loading on intercalated carbon nitride: Enhanced charge interlayer transportation and suppressed backwark reaction for overall water splitting. *Journal of Catalysis* 377, 72-80 (2019).
- He H, et al. Distinctive ternary CdS/ $\text{Ni}_2\text{P}/\text{g-C}_3\text{N}_4$ composite for overall water splitting: Ni_2P accelerating separation of photocarriers. *Applied Catalysis B: Environmental* 249, 246-256 (2019).
- Lin Y, Su W, Wang X, Fu X, Wang X. LaOCl-Coupled Polymeric Carbon Nitride for Overall Water Splitting through a One-Photon Excitation Pathway. *Angew Chem Int Ed Engl* 59, 20919-20923 (2020).
- Liu J, et al. Water splitting. Metal-free efficient photocatalyst for stable visible water splitting via a two-electron pathway. *Science* 347, 970-974 (2015).
- Che W, et al. Fast Photoelectron Transfer in (C_{ring})- C_3N_4 Plane Heterostructural Nanosheets for Overall Water Splitting. *J Am Chem Soc* 139, 3021-3026 (2017).
- Chen X, et al. Three-dimensional porous g- C_3N_4 for highly efficient photocatalytic overall water splitting. *Nano Energy* 59, 644-650 (2019).
- Zhang K, et al. Tunable Bandgap Energy and Promotion of H_2O_2 Oxidation for Overall Water Splitting from Carbon Nitride Nanowire Bundles. *Advanced Energy Materials* 6, (2016).
- Chen L, et al. Graphitic Carbon Nitride Microtubes for Efficient Photocatalytic Overall Water Splitting: The Morphology Derived Electrical Field Enhancement. *ACS Sustainable Chemistry & Engineering* 8, 14386-14396 (2020).
- Zeng Y, et al. Sea-urchin-structure g- C_3N_4 with narrow bandgap (~ 2.0 eV) for efficient overall water splitting under visible light irradiation. *Applied Catalysis B: Environmental* 249, 275-281 (2019).

Q5: The authors did not consider the effect of Pt cocatalyst on the F-CN. It was claimed that the adjacent N site of the C-F structure is responsible for the OER test; what is the reason for the largely enhanced HER half-reaction.

Response: Thanks for your valuable comment. As HER cocatalyst, Pt loading should only affect HER half reaction rather than the CN catalyst itself. We monitored the XPS C 1s, N 1s, and F 1s spectra before and after Pt loading, and no changes were observed (**Fig. R7** and **Fig. R8**), indicating that Pt loading hardly affects the surface C and N atoms in CN/F-CN.

Furthermore, in our experiments, we observed enhanced water splitting efficiency on F-CN compared to CN and CN-E, which should be due to the enhanced OER rather than HER since the four electron-participated OER is the rate-determining

step in water splitting. We further tested the HER efficiencies on CN, CN-E, and F-CN by using triethanolamine as the hole scavenger (**Fig. R9**). The result shows that the HER efficiency on F-CN is not significantly enhanced in comparison with CN and CN-E, demonstrating that the HER half-reaction is not severely affected by the F-modification. The enhanced water splitting efficiency stems from enhanced OER half-reaction. Please see **Page 9, Line 11**.

Fig. R7. The comparison of CN catalysts with and without Pt cocatalyst loading on XPS (a) C 1s, (b) N 1s, and (c) Pt 4f spectra.

Fig. R8. The comparison of F_{0.1}-CN catalysts with and without Pt cocatalyst loading on XPS (a) C 1s, (b) N 1s, (c) F 1s, and (d) Pt 4f spectra.

Fig. R9. The H₂ production profiles on CN, CN-E, and F_{0.1}-CN catalysts with triethanolamine as the hole scavenger.

Q6: The stoichiometric ratio of hydrogen to oxygen production is far from two to one, which is crucial for studying fully decomposed water. The authors need to better illustrate this point by calculating Faraday efficiency or STH. In addition, the author can make a graph or bar chart with the hydrogen/oxygen ratio as the ordinate and the fluorine doping amount as the abscissa to study the influence of fluorine doping.

Response: Thanks for your valuable comment. Due to the inevitable generation of H₂O₂, the UV-induced conversion of O₂

to ozone, and the presence of dissolved O_2 , it is reasonable that the H_2/O_2 ratio for the overall water splitting on semiconductor photocatalyst to be larger than a perfect stoichiometric ratio of 2:1. Similar phenomena were also observed in many other research works (Bai Y, et al, *Angew. Chem. Int. Ed.* 134, e202201299 (2022)). In our system, the main reason for the shortage of O_2 production is the formation of H_2O_2 , since CN is well-acknowledged as an efficient catalyst for H_2O_2 production (Nat. Commun. 12, 3701 (2021); ACS Catalysis 10, 14380-14389 (2020)). To quantify the H_2O_2 production during the reaction, we have employed a Ce^{4+} back titration method (as reported in *Adv. Mater.* 34, e2107480 (2022)) (detailed methods see Supporting Information). As shown in **Fig. R10**, the H_2O_2 production rate on the champion $F_{0.1}$ -CN catalyst was determined to be $85.36 \mu\text{mol}\cdot\text{g}^{-1}\cdot\text{h}^{-1}$ under white light, almost identical to the short of $H_2:O_2$ ratio.

The STH of the whole reaction was also calculated based on the O_2/H_2O_2 ratio (under AM1.5G simulated solar irradiation) on $F_{0.1}$ -CN. According to the following equation, the STH of $F_{0.1}$ -CN was determined to be 0.00195%.

$$\begin{aligned} \text{STH} (\%) &= \frac{R_{H_2} \times \Delta G_T}{P_{sun} \times S} \\ &= \frac{R_{H_2}^1 \times \Delta G_1}{P_{sun} \times S} \times 100\% + \frac{R_{H_2}^3 \times \Delta G_3}{P_{sun} \times S} \times 100\% \\ &= \frac{42.3 \times 0.03 \times 10^{-6} \times 237.1 \times 10^3}{3600 \times 0.3 \times 28.26} \times 100\% + \frac{41.75 \times 0.03 \times 10^{-6} \times 235.3 \times 10^3}{3600 \times 0.3 \times 28.26} \times 100\% \\ &= 0.00195\% \end{aligned}$$

- (1) $2H_2O(l) \rightarrow 2H_2(g) + O_2(g) \quad \Delta G_1 = 237.1 \text{ KJ}\cdot\text{mol}^{-1}$
- (2) $H_2O(l) + \frac{1}{2}O_2(g) \rightarrow H_2O_2(l) \quad \Delta G_2 = 116.7 \text{ KJ}\cdot\text{mol}^{-1}$
- (3) $2H_2O(l) \rightarrow H_2(g) + H_2O_2(l) \quad \Delta G_3 = 235.3 \text{ KJ}\cdot\text{mol}^{-1}$

Moreover, as requested, the bar chart with the H_2/O_2 ratio as functions of the F-content in F-CN samples was depicted as shown in **Fig. R11**. Result shows that the H_2/O_2 ratio was slightly increased with the increasing F-content in F-CN samples, but typically around 3~4. Corresponding descriptions and calculations were added to the revised manuscript. Please see **Page 8, Lines 5-9**.

Fig. R10. (a) UV-vis absorption spectra of Ce^{4+} solution in different concentrations (0.15 mM-0.8 mM); (b) The concentration standard curve of Ce^{4+} solution. (c) The H_2O_2 production profiles on $F_{0.1}$ -CN determined by the back titration of Ce^{4+} .

Fig. R11. The H_2/O_2 evolution rate ratio profile on different F-CN catalysts.

Q7: As far as I know, this journal has high requirements for repeatability of data, especially for statistics figures, such as Fig.2 a (the most important performance data I think); the error bars are usually needed and defined thoroughly.

Response: Thanks for your valuable comment. As requested, error bars were added to Fig. 2a (Fig. R12) and the corresponding bar chart of H₂/O₂ ratios (Supplementary Fig. 17), which was defined by statistically repeating identical experimental results three times. Please see Fig. 2a and Supplementary Fig 17 in the revised manuscript.

Fig. R12. Photocatalytic H₂ and O₂ productions from pure water on pristine CN, CN-E, and different F-CN catalysts under white light illumination. Error bars were obtained by statistically repeating identical experimental results three times.

Q8: The captions in supplementary Figure 5 are inconsistent with the figure curves.

Response: Thanks for your valuable comment. The corresponding description error in the caption of Supplementary Figure 5 (Supplementary Figure 7 in the revised manuscript) was corrected.

To Reviewer 2:

Though efforts have been taken by the authors, part of the comments have not been well responded:

Response: Thanks for your great efforts in reviewing our manuscript. We appreciate your valuable comments and suggestions.

Q1: Q1 ("In Fig. 3c, the rate-determining step of O₂ evolution over CN(C) is the formation of *OH rather than the step from O* to *OOH. So, why the formation of C=O is the bottleneck of overall water splitting on CN-based catalysts? More illustration or an additional attention on the effect of C=O should be given out.") is not properly responded. It seems that they authors do not make clear in the manuscript that what is "the formation of C=O". Is it the same with "the step from *OH to O*"? Even taking consideration of this point, some important even crucial conclusions are inconsistent with the DFT results. It can be seen from Fig. 3c that, the formation energy barrier of *OH is 2.25 eV and the energy barrier of O* to *OOH is about 2 eV while the corresponding value of the formation of O* or C=O is about 0.85, which indicates the rate-determining step of O₂ evolution over CN(C) is the formation of *OH. This conflicts with the authors conclusion that "formation of C=O intermediate is an important bottleneck for overall water splitting on single-phased CN" and even the title "Intermediate C=O Formation Is the Bottleneck of Overall Water Splitting on Carbon Nitride".

Response: Thanks for your valuable comment. Yes, we have made an error when describing the relationship between the C=O intermediate and OER barriers on CN. From the DFT calculations, the step of *O→*OOH (the further transition of C=O, with the break of the double bond) clearly has a larger barrier than the step of *OH→*O (the formation of C=O), which is essentially an important bottleneck of OER on single-phased CN catalysts with observable C=O accumulation during the reaction. We have modified the incorrect description in the revised manuscript; see **Page 9 Line 15**. We assumed that the transition of *O to *OOH is essentially the rate-determining step of OER on CN with C=O accumulation, Thus, the description of "the C=O formation is the bottleneck" is not technically accurate. The title is correspondingly changed to "**Intermediate C=O Transition Is the Bottleneck of Overall Water Splitting on Carbon Nitride**". Thank you again for your very useful suggestions.

Q2: The answer to Q2 (The authors proposed that the N site is the main OER center in F-CN. While N 1s XPS spectra indicates no obvious shift under continuous white light illumination. Why an oxidized N state did not form like C in CN?) is unconvincing. From the results of DFT, the energy barrier of CN(C) and F-CN(N) has a similar trend. So, the statement that "charge transfer at the N site was very fast with no obvious accumulation of intermediate states" lead to the unchanged N 1s state in F-CN is confusing. Please offer a more reasonable explanation.

Response: Thanks for your valuable comment. We think that C atoms have an inert oxidation intermediate state to form C=O, and N atoms have no inert oxidation intermediate states. During the XPS measurements, the OER reaction on CN was continuous. Inert C=O intermediate state can be accumulated by collectively oxidizing C atoms during the reaction, leaving an observable shift on C 1s spectra. However, without an inert intermediate N oxidation state, *O intermediate hardly accumulated on N atoms, which is possibly the reason that N 1s spectra are almost unchanged during reactions.

Q3: The theoretical O₂ production of CN is very low and may exceed the minimum detection value of TCD. The original GC spectra of pristine CN photocatalytic water splitting (e.g., 5h results of Fig. 2b) should be given.

Response: Thanks for your valuable comment. The original GC-TCD spectra of **Fig. 2b** were provided (**Fig. R1**), from which the O₂ production differences between CN and F_{0.1}-CN can be clearly observed.

Fig. R1. The original GC spectra of CN (a) and F_{0.1}-CN (b) samples for photocatalytic water splitting under AM1.5G irradiation.

Q4: For Q4, we advise the authors to clarify the stability of *OH and the adsorption strength of *OH. Is it possible to use electrochemical methanol oxidation experiment to offer further evidence?

Response: Thanks for your valuable comment. Based on the calculated charge density difference mapping, we assume that the F modification in F-CN optimizes the bonding interaction between CN surface and *OH intermediate, which effectively avoids the excessively strong C-O interaction or weak N-O interaction. As carbon sites were occupied by F, leaving less C-OH coordination, therefore the N coordinated *OH on F-CN should be less stable than C coordinated *OH on CN. As suggested, we conducted the electrochemical methanol oxidation reaction (MOR) on CN and F-CN to mimic the break of *OH. As shown in **Fig. R2** and **Fig. R3**, MOR potentials on both samples were lower than the corresponding OER potentials. Particularly, the MOR on F_{0.1}-CN was slightly lower than that on CN, which coincides with the less stable *OH on F-CN. However, the MOR contains the break of both C-H and O-H, and the break of C-H is clearly the high-energy barrier step in MOR. Thus, we think the observed shift of MOR potential majorly reflects the reaction energy of C-H break rather than the absolute stability of *OH on different samples.

Fig. R2. Electrochemical MOR polarization curves on CN, CN-E and F_{0.1}-CN sample.

Fig. R3. Electrochemical OER polarization curves on CN, CN-E and F_{0.1}-CN sample.

Q5: For Q7, please carefully check the reference of CN materials (e.g., Angew. Chem. Int. Ed. 2018, 57, 6848–6852) and get a basic understanding about its structure. Are you sure "there are two types of N one type of C"?

Response: Thanks for your valuable comment. In our first response, we only considered the coordination number of C and N atoms and indicated that there are two types of N coordination (N_{3c} and N_{2c}) and one type of C coordination (C_{3c}). If taking into account the neighboring chemical environments in CN structure, the C site can be further divided into two types, i.e., three-coordinated C_{3c}^1 and three-coordinated C_{3c}^2 (As depicted in **Fig. R4**). F atom should be able to connect with both types of C atoms, and we only considered C_{3c}^1 in our original manuscript. Thus, we further conducted DFT calculation on F-CN with C_{3c}^2 occupied. As shown in **Fig. R5**, the reaction energies of OER on F-CN with occupied C_{3c}^1/C_{3c}^2 and its influence on the neighboring N site are almost identical. The distinguish of C sites and additional DFT calculation results were added to the revised manuscript. Please see **Page 10 Line 12**. Thank you again for your very useful suggestions.

Fig. R4. Simulated structural configuration of CN supercell (2×3).

Fig. R5. (a) Free energy profiles of OER on F-CN at $\text{pH} = 7$ and $U = 0 \text{ V vs SHE}$ (where * represents the intermediate state). C_{3c}^1 profile represents N reaction sites with neighbored C_{3c}^1 site occupied by F atom; C_{3c}^2 profile represents N reaction sites with neighbored C_{3c}^2 site occupied by F atom; (b) PDOS of 2p states of surface C, N and F in F-CN with C_{3c}^2 occupied by F atom. The dashed line stands for Fermi level.

To Reviewer 3:

The revision is satisfactory.

Response: Thanks for your great efforts in reviewing our manuscript.

To Reviewer 4:

In the revised manuscript NCOMMS-22-17621A, the authors highlighted the innovation of this work compared to previous reported similar works, and conducted a systematic comparison of overall water splitting activities with recently reported CN-based catalysts. The authors have added some experimental results (AFM, EIS and Quantification of H₂O₂) to support the structure and reaction mechanism of F-CN according to the reviewers' suggestions. The quality of the manuscript has been improved significantly, and now I recommend the publication of this work in Nature Communication. However, the related experimental results are still controversial. The authors should address the following issues.

Response: Thanks for your great efforts in reviewing our manuscript. We appreciate your valuable comments and suggestions.

Q1: The authors claimed that no changes are found on the N1s spectra on both CN and F_{0.1}-CN samples under the white light illumination. However, the peak area ratios of NH_x, NC₃, C-N=C change significantly. The authors believed that the hybridization of F and C atoms affects the electron orbital of N atoms. This is not sufficient to explain the changes in the in situ NAP-XPS. In addition, detailed results of the XPS fit parameters should be provided.

Response: Thanks for your valuable comment. As shown in **Supplementary Fig. 4 (Fig. R1)**, the difference between CN and F_{0.1}-CN at the initial before light-on is due to the F modification. After the light-on, the peak area ratios of N 1s spectra only changed from the initial to 5 min of light irradiation, which is possibly due to the change of light condition. After 5 min of irradiation, the peak area ratios were almost unchanged. Moreover, detailed fit parameters of NAP-XPS data were provided as shown in **Table. R1**.

Fig. R1. *In-situ* NAP-XPS observations of C1s spectra on (a) pristine CN and (b) F_{0.1}-CN catalysts with 0.2 mbar H₂O vapor pressure using a 300 W Xenon lamp as the white light source in 15 min; and *in-situ* NAP-XPS observation of N1s spectra on (c) pristine CN and (d) F_{0.1}-CN catalysts with 0.2 mbar H₂O vapor pressure using a 300 W Xenon lamp as the white light source in 15 min. The white-light photocatalytic H₂O₂ production profiles on F_{0.1}-CN determined by the back titration of Ce⁴⁺.

Table R1. Summary of detailed NAP-XPS fit parameters.

Sample	C-O (530.1 eV)	O-H (531.3eV)	C=O (532.7eV)	Test conditions
CN	11489.16	10983.41	-	Pristine
CN	10279.90	10348.48	-	Light 0min+0.2mbar H ₂ O
CN	10762	10087.80	9667.29	Light 5min+0.2mbar H ₂ O
CN	10396.01	10073.63	9636.29	Light 10min+0.2mbar H ₂ O
CN	9529.22	8910.69	8687.83	Light 15min+0.2mbar H ₂ O
F _{0.1} -CN	12530.89	12337.01	-	Pristine
F _{0.1} -CN	10940.31	10599.66	-	Light 0min+0.2mbar H ₂ O
F _{0.1} -CN	12850.41	12381.24	-	Light 5min+0.2mbar H ₂ O
F _{0.1} -CN	11156.77	10862.88	-	Light 10min+0.2mbar H ₂ O
F _{0.1} -CN	12067.84	11951.31	-	Light 15min+0.2mbar H ₂ O
Sample	C-N=C (398.3eV)	NC ₃ (399.1eV)	NH _x (400.3eV)	Test conditions

CN	13486.99	12226.83	12073.94	Pristine
CN	12373.09	11354.84	11147.84	Light 0min+0.2mbar H ₂ O
CN	11746.45	11358.30	10975.78	Light 5min+0.2mbar H ₂ O
CN	11608.07	11192.55	10816.01	Light 10min+0.2mbar H ₂ O
CN	11180.82	10802.65	10494.91	Light 15min+0.2mbar H ₂ O
F _{0.1} -CN	14549.46	14068.37	13692.77	Pristine
F _{0.1} -CN	14206.92	13148.05	13106.73	Light 0min+0.2mbar H ₂ O
F _{0.1} -CN	12525.98	11180.03	11649.44	Light 5min+0.2mbar H ₂ O
F _{0.1} -CN	13807.75	13376.22	13047.36	Light 10min+0.2mbar H ₂ O
F _{0.1} -CN	12663.65	12279.95	11900.27	Light 15min+0.2mbar H ₂ O
Specie \ CN	Position (eV)	Integral area	Test conditions	
C-C	284.4	19554.74	Pristine	
C-C	284.5	11939.11	Light 0min+0.2mbar H ₂ O	
C-C	284.7	11242.94	Light 5min+0.2mbar H ₂ O	
C-C	284.8	11847.86	Light 10min+0.2mbar H ₂ O	
C-C	285	13261.09	Light 15min+0.2mbar H ₂ O	
N=C-N	287.7	19645	Pristine	
N=C-N	287.8	12981.81	Light 0min+0.2mbar H ₂ O	
N=C-N	288	12139.18	Light 5min+0.2mbar H ₂ O	
N=C-N	288.1	12787	Light 10min+0.2mbar H ₂ O	
N=C-N	288.3	13895	Light 15min+0.2mbar H ₂ O	
Specie \ F _{0.1} -CN	Position (eV)	Integral area	Test conditions	
C-C	284.6	15848.52	Pristine	
C-C	284.7	12973.31	Light 0min+0.2mbar H ₂ O	
C-C	284.7	16903.55	Light 5min+0.2mbar H ₂ O	
C-C	284.7	15140.01	Light 10min+0.2mbar H ₂ O	
C-C	284.7	14902.10	Light 15min+0.2mbar H ₂ O	
N=C-N	288.1	16681.32	Pristine	
N=C-N	288.1	14008.79	Light 0min+0.2mbar H ₂ O	
N=C-N	288.1	17326.03	Light 5min+0.2mbar H ₂ O	
N=C-N	288.2	16069.74	Light 10min+0.2mbar H ₂ O	
N=C-N	288.2	15889.67	Light 15min+0.2mbar H ₂ O	

Q2: The F_{0.1}-CN catalyst exhibited excellent overall water splitting capacity with H₂ evolution rate of 83.89 $\mu\text{mol}\cdot\text{g}^{-1}\cdot\text{h}^{-1}$ and O₂ evolution rate of 21.15 $\mu\text{mol}\cdot\text{g}^{-1}\cdot\text{h}^{-1}$. The authors employed a Ce⁴⁺ back titration method to quantify the H₂O₂ production during the reaction. The H₂O₂ production rate on the champion F_{0.1}-CN catalyst was determined to be 85.36 $\mu\text{mol}\cdot\text{g}^{-1}\cdot\text{h}^{-1}$. Considering the formation of H₂O₂, it seems that the H₂/O₂ production ratio on the F_{0.1}-CN catalyst would be less than the stoichiometric ratio of 2:1. Why?

Response: Thanks for your valuable comment. The measured H₂O₂ production rate of 85.36 $\mu\text{mol}\cdot\text{g}^{-1}\cdot\text{h}^{-1}$ was obtained under white light irradiation. The mentioned H₂ evolution rate of 83.89 $\mu\text{mol}\cdot\text{g}^{-1}\cdot\text{h}^{-1}$ and O₂ evolution rate of 21.15 $\mu\text{mol}\cdot\text{g}^{-1}\cdot\text{h}^{-1}$ on F_{0.1}-CN were obtained under AM1.5G simulated solar irradiation. The H₂O₂ production under AM1.5G irradiation was measured as 41.75 $\mu\text{mol}\cdot\text{g}^{-1}\cdot\text{h}^{-1}$ (**Fig. R2**), which is consistent with the H₂/O₂ ratio under the identical reaction condition. Please see **Page 8 Line 5**.

Fig. R2. The photocatalytic H₂O₂ production profiles under AM1.5G irradiation on F_{0.1}-CN determined by the back titration of Ce⁴⁺.

Q3: In Supplementary Fig. 9b, the authors should clearly note that the UV-vis absorption of the F-CN catalyst was slightly blue-shifted with high F content.

Response: Thanks for your valuable comment. An additional description of UV-vis absorption data was added to the revised manuscript. Please see **Page 8, Line 30**.

To Reviewer 5:

In this manuscript, the authors used in-situ FTIR and NAP-XPS with isotopic detection to reveal that C=O formation is the inhibiting factor on carbon nitride for photocatalytic overall water splitting to form H₂ and O₂. A surface fluorination strategy was employed to improve the OER performance by occupying carbon-sites on CN with F-ions. This work is insightful and can be published after revision.

Response: Thanks for your great efforts in reviewing our manuscript. We appreciate your valuable comments and suggestions.

Q1: Why the authors choose fluorination strategy to inhibit the formation of C=O? There could be many other choices (such like sulfuration or phosphorization) to occupy the C sites of CN, but the authors seemed to go to fluorination without showing the underlying reason.

Response: Thanks for your valuable comment. We used the fluorination strategy because the chemical binding energy of C-F is very strong. To have a significant effect on the N site, the additional atom needs to have binding energy with C greater than N, and O/F are the only choices. Since the C=O binding is the intermediate state that we wanted to eliminate, we chose the F-modification strategy. The specific reason we used F-modification to inhibit C=O formation was added to the revised manuscript. Please see **Page 5, Line 3**.

Q2: To illustrate the effect of F on enhancing the OER performance, the authors should show more detailed OER test rather than just the LSV. To understand why CN is inert to OER and why F-CN is conducive to OER, the OER kinetics analysis would be much more important and inspiring than simply knowing the onset potential.

Response: Thanks for your valuable comment. We further performed the OER half-reaction with 1 M AgNO₃ as the electron acceptor. As shown in **Fig. R1**, the order of magnitude enhanced O₂ production efficiency can be observed on F_{0.1}-CN in comparison with CN and CN-E. This result significantly demonstrates the enhanced OER ability on F-CN. The additional experimental data was added to the revised manuscript. Please see **Page 9, Line 12**. Thank you again for your very useful suggestion.

Fig. R1. OER profiles on (a) CN and (b) CN-E, and (c) F_{0.1}-CN catalysts with 1 M AgNO₃ as electron acceptor under white light irradiation.

Reviewers' Comments:

Reviewer #1:

Remarks to the Author:

The author responds well to our questions, and the quality of the manuscript has been improved according to the requirements.

Basically, the paper has reached the criterion of this journal for publication.

Reviewer #2:

Remarks to the Author:

The authors have made changes to further improve the manuscript. However, a few crucial points were still evaded.

1、Q1 ("In Fig. 3c, the rate-determining step of O₂ evolution over CN(C) is the formation of *OH rather than the step from O* to *OOH. So, why the formation of C=O is the bottleneck of overall water splitting on CN-based catalysts? ") has been asked twice but the author has not given a direct answer. First, the energy barrier value of each step should be marked clearly in Fig. 3c. As shown in the figure attached, the energy barrier from O* to *OOH is about 1.93 eV, which is obviously smaller than 2.25 eV for the formation of *OH. This is conflict with the assumption that the formation of C=O is the bottleneck.

2、Additional and convincing evidence needs to be provided to support the "essential role" in promoting the intermediate C=O transition when it is not the bottleneck of overall water splitting over C₃N₄ as deduced from DFT results. Otherwise, what's the main contribution of this work and how to make it logical?

3、For the isotope study in DRIFTS, why C=O₁₆ was still present and very strong when CN samples were heated at 423 K for 30 min prior isotopic experiment to exclude H₂O/H₂O₁₈ exchange (supporting information)? Moreover, 201 cm⁻¹ shift was observed after isotopic exchange. The authors are recommended to clarify the reasonability of this huge shift.

Reviewer #4:

Remarks to the Author:

In the revised manuscript of NCOMMS-22-17621B, the authors have clarified the questions raised in the second round of review. Now I recommend the publication of this work in Nature Communication.

Reviewer #5:

Remarks to the Author:

My comments have been well addressed, and the revised version is acceptable.

Reply to reviewers' comments.

To Reviewer 2:

The authors have made changes to further improve the manuscript. However, a few crucial points were still evaded.

Response: Thanks for your great efforts in reviewing our manuscript. We appreciate your valuable comments and suggestions.

Q1: Q1 ("In Fig. 3c, the rate-determining step of O₂ evolution over CN(C) is the formation of *OH rather than the step from O* to *OOH. So, why the formation of C=O is the bottleneck of overall water splitting on CN-based catalysts? ") has been asked twice but the author has not given a direct answer. First, the energy barrier value of each step should be marked clearly in Fig. 3c. As shown in the figure attached, the energy barrier from O* to *OOH is about 1.93 eV, which is obviously smaller than 2.25 eV for the formation of *OH. This is conflict with the assumption that the formation of C=O is the bottleneck.

Response: Thanks for your valuable comment. Indeed, from DFT calculations, the formation of *OH (*→*OH) has a higher barrier than the formation of *OOH (*O→*OOH) on CN(C). Thus, it's not technically accurate to assume the formation of C=O is the bottleneck of OER on CN.

As shown in **Fig. R1** with marked energy barrier value for each step, although the *→*OH has the highest barrier of 2.25 eV, the transition of *O→*OOH also has a very close reaction barrier of 1.90 eV over CN(C) sites. Both are high-energy barrier steps in OER over CN(C). To significantly improve the OER efficiency, lower energy barriers for both steps are necessary. Since *O would be easily formed from *OH once formed due to the low barrier in the step of *OH→*O, the specific manifestation of such reaction barriers at the interface would be an observable accumulation of the solid C=O bonding as a signature of the CN catalyst deactivation due to a high barrier in the further transition of *O to *OOH, which is consistent with our DRIFTS and NAP-XPS experimental observations. Thus, rather than acting as an OER bottleneck, the accumulation of C=O during the reaction is closer to representing an inert CN catalyst surface. However, after the surface F-modification, the accumulation of C=O is directly prevented due to the occupation of C sites. Meanwhile, the adjacent N site is also activated, which significantly reduces the reaction energies of both *→*OH and *O→*OOH on the F-CN(N) site to 0.86 eV and 1.58 eV, respectively, indicating that the fluorination strategy can effectively activate the OER and break through the bottleneck of overall water splitting on CN catalysts.

Our original statement that "C=O is the bottleneck" is imprecise. We have changed the presentation throughout the paper based on the logic mentioned above in the revised manuscript to make it more accurate and avoid further confusion. We also summarized the relative reaction energy of each intermediate state compared to * (0 eV) over CN(C), CN(N), and F-CN(N) sites, as shown in **Table. R1**. Please see **Line 23 of Page 3, Line 9 of Page 6, Line 17 of Page 8, Line 17 of Page 9, Line 10 of Page 10, Line 15 of Page 11, Fig. 3c, and Supplementary Table 5**. The title of the manuscript has also been changed to "**Fluorination Break Through the Bottleneck of Overall Water Splitting on Carbon Nitride.**" Thank you again for your valuable comment.

Fig. R1. Free energy profiles of OER on CN and F-CN at pH = 7 and U = 0 V vs SHE (where * represents the intermediate state). CN(C) represents C reaction sites on the pristine CN; CN(N) represents N reaction sites on the pristine CN; F-CN (N) represents N reaction sites on F-CN (C reaction sites occupied entirely by F atoms).

	* (eV)	*OH (eV)	*O (eV)	*OOH (eV)	*O ₂ (eV)	* (eV)
CN (C)	0	2.25	3.10	5.00	4.03	3.61
CN (N)	0	2.86	2.00	4.24	4.08	3.61
F-CN (N)	0	0.86	1.51	3.09	3.94	3.61

Table. R1. Calculated energies of different intermediate state in respect to the initial * state (0 eV).

Q2: Additional and convincing evidence needs to be provided to support the "essential role" in promoting the intermediate C=O transition when it is not the bottleneck of overall water splitting over C₃N₄, as deduced from DFT results. Otherwise, what's the main contribution of this work and how to make it logical?

Response: Thanks for your valuable comment. Typically, the inert catalytic surface for OER is the bottleneck of overall water splitting on CN catalysts. Both *→*OH (2.25 eV) and *O→*OOH (1.90 eV) are high-energy barrier steps in OER over CN(C). Lower energy barriers for both steps are necessary to break through the bottleneck of overall water splitting on CN catalysts. As we responded in **Q1**, since the further transition of C=O (*O→*OOH) has a high-energy barrier, a small amount of formed C=O would inevitably be accumulated during the reaction, as we observed by DRIFTS and NAP-XPS experiments. Although the transition of C=O is not the highest-barrier OER step, the observable C=O accumulation directly represents a hindered OER in the overall water splitting on CN catalysts as a signature. The simple F-modification occupies C sites, preventing the subsequent oxidation of C atoms from forming C=O and activating the adjacent N site. Especially, the barriers of *→*OH and *O→*OOH on the activated N site are decreased to 0.86 eV and 1.58 eV, respectively, significantly lowering the OER barriers on F-CN. Thus, no observable C=O intermediate accumulation was obtained on F-CN, which prevented a deactivated CN surface, consistent with our experimental observations. As an external manifestation of the inert OER catalytic surface on CN, whether the accumulation of C=O can be observed directly corresponds to the activation/deactivation of CN catalysts for overall water splitting, illustrating the essential role of C=O transition. We have improved the logic and presentation of the revised manuscript accordingly. Thank you again for your helpful comment.

Q3: For the isotope study in DRIFTS, why C=O¹⁶ was still present and very strong when CN samples were heated at 423 K for 30 min prior isotopic experiment to exclude H₂O/H₂O¹⁸ exchange (supporting information)? Moreover, 201 cm⁻¹ shift was observed after isotopic exchange. The authors are recommended to clarify the reasonability of this huge shift.

Response: Thanks for your valuable comment. First, the observable C=O¹⁶ signal in H₂O¹⁸/CN system is clearly from the residue water-induced H₂O/H₂O¹⁸ exchange. Although we conducted the pretreatment on CN samples at 423 K, the applied temperature (423 K) was not high enough to eliminate all surface water species on CN. We have performed a temperature-dependent in-situ DRIFTS experiment (**Fig. R2**) to show the change of surface water species from 293K to 673K. Each profile was collected at the applied temperature after remaining for 30 min. It's observed that the residue water (O-H stretching vibration) continues to lose over 423K, indicating insufficient heating at 423K. However, although the residue water can be eliminated with a rather high temperature, the vibration signal from the CN framework (the band around 1500 cm⁻¹) evolves with a temperature higher than 473 K. To avoid the change of CN structure, we chose 423K to pretreat the CN sample. Thus, the detectable C=O¹⁶ signal is inevitable.

Second, the shift of the IR signal is due to the change of C=O vibration frequency with the O¹⁶/O¹⁸ replacement effect. Under ideal conditions, the IR vibration can approximate the harmonic oscillator motion. According to Hooke's law,

$$\sigma = \frac{1}{2\pi c} \sqrt{\frac{k}{m}},$$

without considering changes in the chemical environment and chemical bond force constant k after the isotope

replacement, when the equivalent mass m increases, the vibration frequency would decrease with a distinct redshift towards lower wavenumbers (*J. Catal.* **178**, 395-407 (1998)). Similar phenomena were also observed in our previous work in terms of Ti⁴⁸-H/D→Ti⁴⁹-H/D replacement (*J. Am. Chem. Soc.* **139**, 2083-2089 (2017)). Under our experimental conditions, no other new peaks were observed except a redshifted C=O peak in the H₂O¹⁸/CN system, which is reasonable to ascribe such a shift to the O¹⁶/O¹⁸ replacement effect. Please see **Line 25 of Page 4**.

Fig. R2. *In-situ* DRIFTS spectra collected by heating the CN sample at different temperatures for 30 min.

Reviewers' Comments:

Reviewer #2:

Remarks to the Author:

The manuscript has been improved according to the comments and suggestions proposed. I am pleased to recommend its publication now.

Reply to reviewers' comments.

To Reviewer 2:

The manuscript has been improved according to the comments and suggestions proposed. I am pleased to recommend its publication now.

Response: Thanks for your great efforts in reviewing our manuscript. We appreciate your valuable comments and suggestions.